# Provable Robust Overfitting Mitigation in Wasserstein Distributionally Robust Optimization

**Shuang Liu, Yihan Wang, Yifan Zhu, Yibo Miao, Xiao-Shan Gao**[*]
State Key Laboratory of Mathematical Sciences
Academy of Mathematics and Systems Science, Chinese Academy of Sciences
Beijing 100190, China
University of Chinese Academy of Sciences, Beijing 100049, China

## Abstract

Wasserstein distributionally robust optimization (WDRO) optimizes against worst-case distributional shifts within a specified uncertainty set, leading to enhanced generalization on unseen adversarial examples, compared to standard adversarial training which focuses on pointwise adversarial perturbations. However, WDRO still suffers fundamentally from the robust overfitting problem, as it does not consider statistical error. We address this gap by proposing a novel robust optimization framework under a new uncertainty set for adversarial noise via Wasserstein distance and statistical error via Kullback-Leibler divergence, called the Statistically Robust WDRO. We establish a robust generalization bound for the new optimization framework, implying that out-of-distribution adversarial performance is at least as good as the statistically robust training loss with high probability. Furthermore, we derive conditions under which Stackelberg and Nash equilibria exist between the learner and the adversary, giving an optimal robust model in certain sense.Finally, through extensive experiments, we demonstrate that our method significantly mitigates robust overfitting and enhances robustness within the framework of WDRO.

## 1 Introduction

Optimization in machine learning is often challenged by data ambiguity caused by natural noise, adversarial attacks (Goodfellow et al., 2014), or other distributional shifts (Malinin et al., 2021; 2022). To develop robust models against data ambiguity, Wasserstein Distributionally Robust Optimization (WDRO) (Kuhn et al., 2019) has emerged as a powerful modeling framework, which has recently attracted significant attention attributed to its connections to generalization and robustness (Lee & Raginsky, 2018; Mohajerin Esfahani & Kuhn, 2018; An & Gao, 2021).

Specifically, WDRO optimizes performance under the most adversarial data distribution within a certain Wasserstein distance from the observed empirical distribution $\mathcal{D}_n$, as shown below.

$$\inf_{\theta \in \Theta} \sup_{\mathcal{D}' : \mathcal{D}' \in \mathcal{U}(\mathcal{D}_n)} \mathbb{E}_{z \sim \mathcal{D}'}[L(\theta, z)]. \tag{1}$$

Here $\mathcal{U}(\mathcal{D}_n) = \{\mathcal{D}' : \mathrm{W}_p(\mathcal{D}_n, \mathcal{D}') \leq \varepsilon\}$ is the ambiguity set consisting of all possible distributions of interest, and $L(\cdot, \cdot)$ is the loss function. From both intuitive and theoretical perspectives, WDRO can be considered as a more comprehensive framework that subsumes and extends adversarial training (Staib & Jegelka, 2017; Sinha et al., 2017; Bui et al., 2022; Phan et al., 2023). In contrast to standard adversarial training (Madry et al., 2017), WDRO operates on a broader scale by perturbing the entire data distribution rather than pointwise adversarial samples. This approach inherently promotes generalization to unseen adversarial examples.

However, we find that WDRO still fundamentally suffers from the same robust overfitting phenomenon as standard adversarial training (Rice et al., 2020). This phenomenon observed in our

---

[*]Corresponding author.

experiments as a degradation in adversarial robustness on test data after a certain point in the training process, which typically occurs shortly after the first learning rate decay (refer to Figure 2).

A primary factor contributing to overfitting behavior is the inherent statistical error arising from finite sampling of train data (Van Parys et al., 2021; Bennouna & Van Parys, 2022). This error, defined as the discrepancy between the empirical distribution of the training data and the true underlying data distribution, manifests as a gap between empirical and population risk. Consequently, models optimized via traditional empirical risk minimization (ERM) may perform well on training data but generalize poorly to unseen samples. Although the recent literature Lam (2019); Van Parys et al. (2021); Bennouna & Van Parys (2022) has acknowledged the significance of statistic error in robust optimization frameworks, its incorporation into WDRO remains an unexplored avenue.

To address these issues, we propose a novel ambiguity set obtained by combining Kullback-Leibler divergence and Wasserstein distance, which is called **Statistically Robust WDRO (SR-WDRO)**. We prove that the SR-WDRO formulation guarantee an upper bound on the adversarial test loss for any desired robustness and any loss function. Furthermore, we formalize the SR-WDRO as a zero-sum game between the learner (decision maker), who chooses a model parameter $\theta \in \Theta$, and the adversary who chooses a distribution from an ambiguity set, and we derive conditions under which Stackelberg and Nash equilibria exist between the learner and the adversary.

Finally, building upon our theoretical framework, we introduce a novel and practical statistically robust loss training algorithm for neural networks, which incurs negligible additional computational burden compared to standard adversarial training or distributionally robust training. We conduct extensive experiments on benchmark datasets, which show that our proposed approach demonstrates superior efficacy in mitigating overfitting compared to existing training methodologies, in that our approach maintains performance on in-distribution data while substantially improving robustness to out-of-distribution samples, thus mitigating the critical challenge in generalization.

The theoretical and practical contributions of this paper are summarized as follows.

- We propose SR-WDRO based on our novel ambiguity set to mitigate robust overfitting in WDRO.

- Theoretically, we prove that the adversarial test loss can be upper bounded by the statistically robust training loss with high probability, which can be considered as generalization bound for SR-WDRO. We also establish the conditions necessary for the existence of Stackelberg and Nash equilibria between the learner and the adversary.

- We introduce a practically robust training algorithm based on SR-WDRO which maintains computational efficiency.

- We conduct extensive evaluations on benchmark datasets and our results show that the SR-WDRO approach effectively mitigates robust overfitting and outperforms other existing robust methods in terms of adversarial robustness.

## 2 RELATED WORKS

**Robust Overfitting.** The phenomenon of robust overfitting has been a focal point in the field of adversarial machine learning. The seminal work by Rice et al. (2020) brought this issue to the forefront, demonstrating that after a certain point in standard adversarial training, i.e., shortly after the first learning rate decay, the robust performance on test data will continue to degrade with further training. Notably, conventional overfitting remedies such as explicit regularization and data augmentation do not improve performance beyond early stopping. Schmidt et al. (2018) attributed robust overfitting to sample complexity theory and suggested that more training data are required for adversarial robust generalization. This assertion is supported by empirical results in subsequent studies (Schmidt et al., 2018; Zhai et al., 2019). Recent works also proposed various strategies to mitigate robust overfitting without relying on additional training data by sample re-weighting (Zhang et al., 2020; Wang et al., 2019), adversarial weight perturbation (Wu et al., 2020), weight smoothing (Chen et al., 2020), favoring large loss data (Yu et al., 2022), and considering the memorization effect (Dong et al., 2022; Wang et al., 2024; Yu et al., 2024). In our work, we will introduce the statistically robustness into WDRO to mitigate overfitting theoretically and practically.

**WDRO.** WDRO has recently received significant attention in the context of adversarial training (Staib & Jegelka, 2017; Wong et al., 2019; Sinha et al., 2017; Bui et al., 2022; Phan et al., 2023). Unlike standard adversarial training (Madry et al., 2017), which defends against attacks by bounding the perturbation to each individual data point, WDRO provides protection by imposing a bound on the average perturbation constrained by Wasserstein distance applied on the entire data distribution. This distinction positions WDRO as a more holistic approach to handling adversarial perturbations and generalizes better than adversarial training on unseen data samples (Bui et al., 2022). However, despite this advantage, WDRO remains vulnerable to the same robust overfitting phenomenon observed in standard adversarial training, primarily due to its lack of consideration for statistical error. Recently, Bennouna & Van Parys (2022); Bennouna et al. (2023) proposed a DRO formulation using an ambiguity set combining Kullback-Leibler divergence and Levy-Prokhorov metric in an attempt to protect against corruption and statistical errors. In the absence of data poisoning, their framework effectively reduces to improving standard adversarial training with statistical robustness. However, our approach diverges significantly. We leverage the inherent advantages of WDRO as our foundation and then augment it with statistical robustness to mitigate robust overfitting, which enables our framework to achieve superior performance against adversarial perturbations.

## 3 PRELIMINARIES

**Notations.** Let $\mathcal{Z} \subset \mathbb{R}^m$ be a compact nonempty set equipped with metric $\mathrm{d}(\cdot, \cdot)$ and $\mathrm{diam}(\mathcal{Z}) = \sup\{\mathrm{d}(z, z') : z, z' \in \mathcal{Z}\}$ the diameter of $\mathcal{Z}$ which is finite. The class of Borel probability measures on $\mathcal{Z}$ is denoted by $\mathcal{P}(\mathcal{Z})$. For simplicity, we denote $\mathbb{E}_\mu[f(z)]$ as the expectation of $f(z)$ with $z \sim \mu$.

**Definition 1** (Wasserstein metric). *For $p \geq 1$, the $p$-th Wasserstein metric between $\mu, \nu \in \mathcal{P}(\mathcal{Z})$ is*

$$\mathrm{W}_p(\mu, \nu) := \inf_{\pi \in \Pi(\mu,\nu)} \left\{ \mathbb{E}_{(z,z') \sim \pi} \left[ \mathrm{d}^p(z, z') \right] \right\}^{\frac{1}{p}}$$

*where the infimum is taken over all coupling of $\mu$ and $\nu$, i.e. probability measure $\pi$ on the product space $\mathcal{Z} \times \mathcal{Z}$ with given marginals $\mu$ and $\nu$.*

**Definition 2** (KL divergence). *Kullback-Leibler (KL) divergence between $\mu, \nu \in \mathcal{P}(\mathcal{Z})$ is*

$$\mathrm{KL}(\mu, \nu) := \int_{\mathcal{Z}} p(z) \log \frac{p(z)}{q(z)} dz$$

*where $p(z), q(z)$ are probability density functions of $\mu, \nu$, respectively.*

Wasserstein metric is a function defined between two probability distributions, which represents the cost of an optimal mass transportation plan. Kullback-Leibler divergence measures the difference between two probability distributions, quantifying how one distribution diverges from a reference distribution. We additionally consider the Levy-Prokhorov (LP) metric and the total variation metric (TV). The Levy-Prokhorov metric is theoretically important because weak convergence of probability distribution is equivalent to the convergence in the Levy-Prokhorov metric. In Appendix A.1, we establish the relationship among these probability discrepancies.

## 4 STATISTICALLY ROBUST WDRO

In Section 4.1, we first present the definition and game-theoretic description of SR-WDRO. Subsequently, in Section 4.2, we demonstrate that models trained with statistically robust loss achieve out-of-distribution generalization. Finally, in Section 4.3 we examine the existence of Stackelberg and Nash equilibria under some assumptions.

### 4.1 STATISTICALLY ROBUST WDRO AND GAME DESCRIPTION

We perform stochastic optimization with respect to an unknown data distribution $\mathcal{D}$, given access only to an empirical distribution $\mathcal{D}_n$, where $n$ is the observed finite sample size. The goal of distributionally robust optimization is to learn a robust model from $\mathcal{D}_n$ that will perform well on unseen test distribution $\mathcal{D}_{\text{test}}$, which may either be the true data distribution $\mathcal{D}$ or more likely distributions shifted from $\mathcal{D}$. Our work focuses primarily on distribution shifts caused by adversarial attacks.

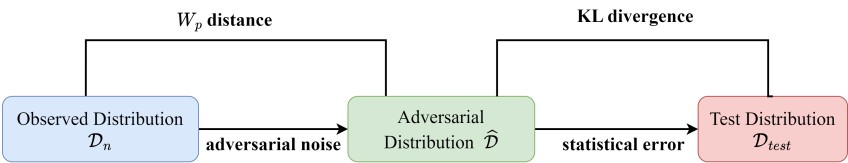

Figure 1: Illustration of our SR-WDRO. The adversarial perturbations are quantified using Wasserstein distance between $\mathcal{D}_n$ and $\widehat{\mathcal{D}}$. The adversarial distribution is then compared with the test distribution $\mathcal{D}_{\text{test}}$ using Kullback-Leibler divergence, which accounts for the statistical error.

The WDRO constructs an ambiguity set based on the empirical distribution $\mathcal{D}_n$, and optimizes performance under the most adversarial distribution within a certain Wasserstein distance from $\mathcal{D}_n$ as shown in Eq (1). However, previous WDRO methods still suffer fundamentally from the robust overfitting phenomenon. To mitigate this issue, we incorporate the Kullback-Leibler (KL) divergence in WDRO, specifically aiming to reduce statistical error caused by training on finite samples, a brief illustration is shown in Figure 1.

Our SR-WDRO can be interpreted as a novel combination of Kullback-Leibler divergence and Wasserstein distributional robust optimizations. We consider the ambiguity set:

$$\mathcal{U}(\mathcal{D}_n) := \{\mathcal{D}' \in \mathcal{P}(\mathcal{Z}) : \exists \mathcal{D}'' \in \mathcal{P}(\mathcal{Z}) \text{ s.t. } W_p(\mathcal{D}_n, \mathcal{D}'') \leq \varepsilon, \text{ KL}(\mathcal{D}'', \mathcal{D}') \leq \gamma\} \tag{2}$$

where the desired protection against adversarial noise is controlled by the *adversarial budget* $\varepsilon$ and statistical error is accounted by the *statistical budget* $\gamma (\gamma > 0)$.

Now we can formulate this robust optimization problem as a *zero-sum game* between the learner who chooses a decision model $\theta \in \Theta$ and an adversary who chooses a distribution $\mathcal{D}'$ from the ambiguity set $\mathcal{U}(\mathcal{D}_n)$. Let $L(\theta, z)$ be the loss function (e.g. cross-entropy loss). Our game model is as follows:

- The adversary chooses a distribution $\mathcal{D}' \in \mathcal{U}(\mathcal{D}_n)$ to maximize the loss $\mathbb{E}_{\mathcal{D}'}[L(\theta, z)]$.
- The learner selects a decision model $\theta \in \Theta$, aiming to minimize the loss $\mathbb{E}_{\mathcal{D}'}[L(\theta, z)]$.

We introduce the *statistically robust WDRO (SR-WDRO) problem*:

$$\inf_{\theta \in \Theta} \sup_{\mathcal{D}' \in \mathcal{U}(\mathcal{D}_n)} \mathbb{E}_{\mathcal{D}'}[L(\theta, z)] \tag{3}$$

and denote the *statistically robust loss* as:

$$\mathcal{L}_{\varepsilon, \gamma}(\theta, \mathcal{D}_n) := \sup_{\mathcal{D}' \in \mathcal{U}(\mathcal{D}_n)} \mathbb{E}_{\mathcal{D}'}[L(\theta, z)]. \tag{4}$$

### 4.2 CERTIFIED GENERALIZATION

In this subsection, we rigorously establish that the model trained with statistically robust loss $\mathcal{L}_{\varepsilon, r}(\theta, \mathcal{D}_n)$ enjoys the asymptotic out of distribution generalization guarantee and safeguards against the statistical error and adversarial noises simultaneously. Let $\mathbb{B}(z; \epsilon) := \{z' \in \mathcal{Z} \mid d(z, z') \leq \varepsilon\}$ be the $\epsilon$-ball around $z$ under metric d. Proofs are provided in Appendix A.2.

**Theorem 3** (Generalization certificate). *Let $\mathcal{D}$ be the true data distribution, and $\mathcal{D}_n$ be the observed empirical distribution sampled i.i.d. from $\mathcal{D}$. Then for all $\varepsilon > 0$, let $\delta = (\frac{\varepsilon}{\text{diam}(\mathcal{Z})+1})^p$, we have*

$$\mathbb{P}\Big(\forall \theta \in \Theta, \mathbb{E}_{\mathcal{D}}[L(\theta, z)] \leq \mathcal{L}_{\varepsilon, \gamma}(\theta, \mathcal{D}_n)\Big) \geq 1 - e^{-\gamma n}\left(\frac{4}{\delta}\right)^{m(\mathcal{Z}, \delta)} \tag{5}$$

*where $m(\mathcal{Z}, \delta) := \min\{k \geq 0 : \exists \xi_1, \cdots, \xi_k \in \mathcal{Z}, \text{ s.t. } \cup_{i=1}^k \mathbb{B}(\xi_i, \delta) \supseteq \mathcal{Z}\}$ is the* internal covering number *of the support set $\mathcal{Z}$.*

In the proof of Theorem 3, we in fact prove that the true distribution $\mathcal{D}$ can be in the ambiguity set $\mathcal{U}(\mathcal{D}_n)$ with high probability. Compared to the results in (Bennouna & Van Parys, 2022), the bound we derived elucidates the dependency on parameters $\varepsilon$ and $\gamma$, and the internal covering number is directly related to $\varepsilon$. Unlike classical statistical learning, our bounds do not depend on the dimension of the parameter space $\Theta$ but rather on the internal covering number of sample space $\mathcal{Z}$. The result is uniform in that the probability approaches 1 for any $\theta$ when the sample size $n$ is sufficiently large.

**Remark 4.** *To ensure that the bound in equation 5 is meaningful, the condition $\gamma > m(\mathcal{Z}, \delta) \cdot \log(4/\delta)/n$ must hold. Although this condition may seem strong due to the exponential dependency on the dimension of the sample space $\mathcal{Z}$, the cover number is not excessively large in practice because the intrinsic dimensionality of the sample space is often much lower than its ambient dimension. For example, the intrinsic dimensionality of MNIST is around 13, CIFAR-10 approximately 26, and ImageNet between 26 and 43 (Facco et al., 2017; Pope et al., 2021). Our theory can leverage this intrinsic dimensionality instead of the ambient dimension, making the condition more practical.*

We extend our analysis to consider adversarial robustness, wherein the test distribution $\mathcal{D}$ is subject to adversarial perturbations. We formalize this scenario by introducing an adversarial transformation:

$$\mathcal{M} : z \to \arg\max_{z' \in \mathbb{B}(z;\epsilon)} L(\theta, z').$$

This transformation maps each test input to its worst-case adversarial example within the $\epsilon$-neighborhood. Let $\mathcal{M}_\#\mathcal{D}$ denote the pushforward measure of $\mathcal{D}$ under $\mathcal{M}$, representing the distribution of adversarial examples generated from $\mathcal{D}$. And the expected loss under this adversarial distribution can be expressed as: $\mathbb{E}_{\mathcal{M}_\#\mathcal{D}} L(\theta, z) = \mathbb{E}_{\mathcal{D}}[\max_{z' \in \mathbb{B}(z;\epsilon)} L(\theta, z')]$. Our main result establishes a high-probability bound on the test adversarial loss:

**Theorem 5** (Robustness certificate). *Let $\mathcal{D}$ be the true data distribution, $\mathcal{D}_n$ the empirical distribution sampled i.i.d. from $\mathcal{D}$, and $\delta = (\frac{\sigma}{\mathrm{diam}(\mathcal{Z})+1})^p$ for any $\sigma > 0$. Then*

$$\mathbb{P}\left(\forall \theta \in \Theta, \mathbb{E}_{\mathcal{D}}[\max_{z' \in \mathbb{B}(z;\epsilon)} L(\theta, z')] \leq \mathcal{L}_{\varepsilon+\sigma,\gamma}(\theta, \mathcal{D}_n)\right) \geq 1 - e^{-n\gamma}\left(\frac{4}{\delta}\right)^{m(\mathcal{Z},\delta)} \tag{6}$$

*where $m(\mathcal{Z}, \delta) := \min\{k \geq 0 : \exists \xi_1, \cdots, \xi_k \in \mathcal{Z}, \text{ s.t. } \cup_{i=1}^{k} \mathbb{B}(\xi_i, \delta) \supseteq \mathcal{Z}\}$ denote the internal covering number of the support set $\mathcal{Z}$.*

From Theorem 5, we observe that ensuring $\varepsilon$-robustness on the test set typically needs a larger attack budget $\varepsilon + \sigma$ during training. However, according to Theorem 3, an excessively large training budget may impede generalization for benign data. These findings align with previous experimental results on adversarial training (Tsipras et al., 2018; Andriushchenko & Flammarion, 2020).

**Remark 6** (Comparison with Standard WDRO). *From a theoretical perspective, our framework differs from standard WDRO in both the assumptions required and the form of the generalization bounds. Specifically, the generalization bounds in (Azizian et al., 2024) rely on additional assumptions about the loss function, such as Lipschitz continuity and boundedness, which are not required in our framework. Furthermore, the form of the generalization bounds in our framework is fundamentally different from those of WDRO. Our bound provides a stronger guarantee, where the probability of failure (i.e., the complement of the confidence level) decays exponentially with the sample size $n$. Importantly, this rate of decay is directly influenced by $\gamma$.*

**Remark 7.** *The result is also applicable to general out-of-distribution shifts such as domain generalization. Details are given in Appendix A.3.*

### 4.3 NASH EQUILIBRIUM AND STACKELBERG EQUILIBRIUM

As mentioned in Section 4.1, the SR-WDRO problem (3) constitutes a zero-sum game. In this subsection, we show that Stackelberg equilibrium exists under natural assumptions and Nash equilibrium exists under more assumptions. The detailed definitions of the game equilibrium, along with the proofs for this section, are provided in the Appendix A.4.

We need the following assumptions on the loss function.

**Assumption 8** (Loss function).

i) *For any $\theta \in \Theta$, the loss function $L(\theta, z)$ is non-negative and upper semi-continuous in $z$.*

ii) *For any $z \in \mathcal{Z}$, the loss function $L(\theta, z)$ is lower semi-continuous in $\theta$.*

iii) *For any $\theta \in \Theta$, there exist $c > 0$, $\widehat{z}_0 \in \mathcal{Z}$ and $k \in (0, p)$ such that $L(\theta, z) \leq c[1 + \mathrm{d}^k(z, \widehat{z}_0)]$ for all $z \in \mathcal{Z}$.*

Assumptions (i) and (ii) regarding the continuity of the loss function are common and are satisfied by neural network models, and assumption (iii) is necessary for the existence of the maximum in (4). The following proposition establishes that the inner maximization problem is solvable, which is the key ingredient required to prove the existence of Stackelberg and Nash equilibrium.

**Proposition 9.** *The ambiguity set $\mathcal{U}(\mathcal{D}_n) := \{\mathcal{D}' \in \mathcal{P}(\mathcal{Z}) : \exists \mathcal{D}'' \in \mathcal{P}(\mathcal{Z}) \text{ s.t. } \mathrm{W}_p(\mathcal{D}_n, \mathcal{D}'') \leq \varepsilon, \mathrm{KL}(\mathcal{D}'', \mathcal{D}') \leq \gamma\}$ is compact in $\mathcal{P}(\mathcal{Z})$ with respect to weak topology. If Assumption 8 is satisfied, then the supremum in (4) is finite and can be attained for any $\theta \in \Theta$.*

**Stackelberg game:** We also consider a threat model in which the adversary has full knowledge of the learner's actions before determining its own strategy. We can model our SR-WDRO problem (3) as a zero-sum Stackelberg game between the leader (decision-maker) who chooses a decision model $\theta \in \Theta$ at first, and the follower (adversary) who chooses a distribution $\mathcal{D}'$ from the ambiguity set $\mathcal{U}(\mathcal{D}_n)$ after observing the leader's action. As a corollary of Proposition 9, we can give the existence of Stackelberg equilibrium.

**Theorem 10** (Stackelberg Equilibrium). *If Assumption 8 holds and $\Theta$ is compact, then the game has a Stackelberg equilibrium. i.e. there exists a $(\theta^*, \mathcal{D}'^*(\theta^*))$ such that*

$$\theta^* \in \underset{\theta \in \Theta}{\arg\min} \max_{\mathcal{D}' \in \mathrm{BR}(\mathcal{D}_n)} \mathbb{E}_{\mathcal{D}'}[L(\theta, z)], \quad \mathcal{D}'^*(\theta^*) \in \mathrm{BR}(\mathcal{D}_n) = \underset{\mathcal{D}' \in \mathcal{U}(\mathcal{D}_n)}{\arg\max} \mathbb{E}_{\mathcal{D}'}[\mathrm{L}(\theta^*, z)].$$

In addition, with the convexity of $L(\theta, z)$ in $\theta$, we can prove the existence of Nash equilibrium.

**Theorem 11** (Minimax Theorem). *If Assumption 8 holds, $\Theta$ is convex, and $L(\theta, z)$ is convex in $\theta$ for any $z \in \mathcal{Z}$, then*

$$\min_{\theta \in \Theta} \max_{\mathcal{D}' \in \mathcal{U}(\mathcal{D}_n)} \mathbb{E}_{\mathcal{D}'}[L(\theta, z)] = \max_{\mathcal{D}' \in \mathcal{U}(\mathcal{D}_n)} \min_{\theta \in \Theta} \mathbb{E}_{\mathcal{D}'}[L(\theta, z)]. \tag{7}$$

We thus have established the existence for the Stackelberg and Nash equilibria of the SR-WDRO problem (3), viewed as a zero-sum game between the learner and the adversary. The existence of Stackelberg equilibrium is considered for the first time for WDRO approaches. It can be observed that the Stackelberg equilibrium requires weaker conditions to exist than the Nash equilibrium, which gives the *smallest statistically robust loss* among all decision models can be considered optimal robust in certain sense. Our result significantly extends the scope of WDRO theory and tackles a substantially more complex and general formulation compared with previous work (Shafieezadeh-Abadeh et al., 2023).

## 5 COMPUTATIONALLY TRACTABLE REFORMULATION OF SR-WDRO

Having established the robustness certificate and equilibrium existence of SR-WDRO, we now turn our attention to practical methods for solving the optimization problem defined in Equation (3). We first show that the inner maximization problem (4) can be simplified to a minimization problem. Next, we apply our previous results to classification tasks and provide an approximate algorithm to solve the robust loss effectively.

**Proposition 12** (Strong duality). *$\mathcal{L}_{\varepsilon,\gamma}(\theta, \mathcal{D}_n)$ in (4) admits the dual formulation for all $\gamma > 0$:*

$$\mathcal{L}_{\varepsilon,\gamma}(\theta, \mathcal{D}_n) = \inf_{\substack{\lambda, \beta \geq 0 \\ \eta \geq \max_{z \in \mathcal{Z}} L(\theta, z)}} \{\lambda \varepsilon^p + \mathbb{E}_{\mathcal{D}_n}[\varphi(\lambda, \beta, \eta, z)]\} \tag{8}$$

*where $\varphi(\lambda, \beta, \eta, z) = \sup_{\xi \in \mathcal{Z}} \{\beta \log \frac{\beta}{\eta - L(\theta, \xi)} + (\gamma - 1)\beta + \eta - \lambda \mathrm{d}^p(z, \xi)\}$.*

In contrast, the dual reformulation for classical WDRO only involves $\lambda$ and takes the expectation of the implicit function $\sup_{\xi \in \mathcal{Z}} \{L(\theta, \xi) - \lambda \mathrm{d}^p(z, \xi)\}$ with respect to $\mathcal{D}_n$. The additional variables $\beta, \eta$ are introduced to account for the KL divergence.

---

**Algorithm 1** Statistically Robust WDRO Training

---

**Input**: Training set $\mathcal{S}_n$, number of iterations $T$, batch size $N$, learning rate $\eta_\theta$, $\eta_\lambda$, adversary parameters: attack budget $\varepsilon$, steps $K$, step size $\eta$.
**Output**: Robust model $\theta$.

  1: **for** $t = 1$ to $T$ **do**
  2:     Sample a mini-batch $\{(x_i, y_i)\}_{i=1}^N \in \mathcal{S}_n$
  3:     Find UDR adversarial examples $\{x_i^a\}_{i=1}^N$:
  4:        (1) $x_i^0 = x_i + \delta$ where $\delta \sim \text{Uniform}(-\varepsilon, \varepsilon)$
  5:        (2) for $k = 1$ to $K$:
  6:           (a) $x_i^{inter} = x_i^{k-1} + \eta \text{sign}(\nabla_x L(\theta, (x_i^{k-1}, y_i)))$
  7:           (b) $x_i^k = x_i^{inter} - \eta\lambda\nabla_x \widehat{d}(x_i^{inter}, x_i)$
  8:        (3) Clip to valid range: $x_i^a = \text{clip}(x_i^K, 0, 1)$
  9:     Update parameter $\lambda$: $\lambda \leftarrow \lambda - \eta_\lambda \left(\varepsilon - \frac{1}{N}\sum_{i=1}^N \widehat{d}_{\mathcal{X}}(x_i^a, x_i)\right)$
10:     Compute optimal weights $\{p_i\}_{i=1}^N$ in problem (9)
11:     Update model parameter: $\theta \leftarrow \theta - \eta_\theta \nabla_\theta \mathcal{L}_{\varepsilon,r}(\theta; \{(x_i, y_i)\}_{i=1}^n)$
12: **end for**

---

However, directly solving the statistically robust loss through either (4) or (8) is intractable in practice. In the remainder of this section, we will primarily focus on applying results to classification tasks and provide a finite reformulation to solve the statistically robust loss approximately.

In classification tasks, it is often intuitive to restrict adversarial perturbations solely to input samples (feature vectors). We present an adaptation of existing results to such scenarios. Let $\mathcal{Z} = (X, Y) \in \mathcal{X} \times \mathbb{R}$, where $X \in \mathcal{X} \subset \mathbb{R}^{(m-1)}$ represents input data, and $Y \in \mathbb{R}_+$ denotes a label. In the classification settings, $Y \in \{1, ..., K\}$. We consider an adversary capable of perturbing only the input samples $X$. This constraint can be elegantly incorporated into our robust formulation by defining the Wasserstein cost function $d : \mathcal{Z} \times \mathcal{Z} \to \mathbb{R}+$ as follows. For $z = (x, y)$ and $z' = (x', y')$, we introduce the *sample-shift cost function*:

$$\text{d}(z, z') = \text{d}_{\mathcal{X}}(x, x') + M \cdot \mathbf{1}\{y \neq y'\}$$

where $\text{d}_{\mathcal{X}}$ is the distance (like $L_q$ norm) on $\mathcal{X}$ and we assume that $M$ is a very large positive number to prevent label conversion.

Now we revisit the statistically robust loss:

$$\mathcal{L}_{\varepsilon,\gamma}(\theta, \mathcal{D}_n) = \sup_{\mathcal{D}' \in \mathcal{U}(\mathcal{D}_n)} \mathbb{E}_{\mathcal{D}'}[L(\theta, z)]$$
$$= \sup\left\{\sup\left\{\mathbb{E}_{\mathcal{D}'}[L(\theta, z)] : \mathcal{D}', \text{KL}(\mathcal{Q}, \mathcal{D}') \leq \gamma\right\} : \mathcal{Q}, \text{W}_p(\mathcal{D}_n, \mathcal{Q}) \leq \varepsilon\right\}.$$

Intuitively, we start with the empirical distribution $\mathcal{D}_n$ and identify the most adversarial sample distribution that satisfies the Wasserstein distance constraint. Then, within the KL divergence constraint, similarly to (Bennouna et al., 2023), we adjust the sample weights to find the sample distribution that maximizes the final loss. To identify the most adversarial sample distribution, we employ the UDR method (Bui et al., 2022), which leverages the dual formulation of WDRO. In order to match this method, we set $p = 1$ in our framework. This approach constructs a new Wasserstein cost function to search for adversarial samples that incorporate both local and global information, in contrast to Adversarial Training that considers only local information for each sample.

For any given training data (sub)set $\{z_i = (x_i, y_i)\}_{i=1}^n$, we can compute the statistically robust loss $\mathcal{L}_{\varepsilon,\gamma}(\theta)$ approximately as

$$\mathcal{L}_{\varepsilon,\gamma}(\theta, \{(x_i, y_i)\}_{i=1}^n) = \begin{cases} \max \sum_{i=1}^n p_i L(\theta, (x_i', y_i)) \\ \quad\quad x_i' \in \arg\max_{x_i'} L(\theta, (x_i', y_i)) - \lambda\widehat{d}_{\mathcal{X}}(x_i', x_i) \\ \text{s.t.} \quad \boldsymbol{p} \in \mathbb{R}_+^n, \sum_{i=1}^n p_i = 1, \\ \quad\quad \boldsymbol{q} \in \mathbb{R}_+^n, q_i = \frac{1}{n} \,\forall i = 1, \cdots, n, \\ \quad\quad \text{KL}(\boldsymbol{q}||\boldsymbol{p}) = \sum_{i=1}^n q_i \log\left(\frac{q_i}{p_i}\right) \leq \gamma. \end{cases} \quad (9)$$

The $\lambda$ comes from the dual formulation of WDRO and is a learnable parameter in UDR, and

$$\widehat{d}_{\mathcal{X}}\left(x, x'\right) := \mathbf{1}\left\{\mathrm{d}_{\mathcal{X}}\left(x, x'\right) < \varepsilon\right\}\mathrm{d}_{\mathcal{X}}\left(x, x'\right) + \mathbf{1}\left\{\mathrm{d}_{\mathcal{X}}\left(x, x'\right) \geq \varepsilon\right\}\left(\varepsilon + \frac{\mathrm{d}_{\mathcal{X}}\left(x, x'\right) - \varepsilon}{\tau}\right)$$

where $\tau > 0$ is the temperature to control the growing rate of the cost function when $x'$ is out of the perturbation ball.

The statistically robust loss is therefore in essence simply a re-weighting of adversarial loss. Determining the statistically robust loss exactly requires evaluating the adversarial loss $L(\theta, z')$ where the adversarial example is computed through the WDRO method (Bui et al., 2022), subsequently solving the exponential cone problem with $n$ variables and constraints. This optimization can be efficiently executed using standard solvers. More details are shown in Algorithm 1.

## 6 EXPERIMENTS

In this section, we investigate the efficacy of our SR-WDRO training through extensive experiments on the CIFAR-10 and CIFAR-100 datasets. We will show that our approach largely circumvents the robust overfitting phenomenon experienced by WDRO and achieves better adversarial robustness than other robust methods. The implementation of our approach is publicly available at the following GitHub repository: https://github.com/hong-xian/SR-WDRO.

We compare our method with PGD-AT (Madry et al., 2017) and distributional robust methods: UDR-AT (Bui et al., 2022), HR training (Bennouna et al., 2023). We train ResNet-18 (He et al., 2016) with 200 epochs, and use SGD as the optimizer with learning rate decay by 0.1 at the epoch 100 and 150. For all methods, we implement adversarial training with $\{k = 10, \varepsilon = 8/255, \eta = 2/255\}$ where $k$ is the iteration number, $\varepsilon$ is the attack budget and $\eta$ is the step size. We use different attacks to evaluate the defense methods, including: 1) PGD-10 with $\{k = 10, \varepsilon = 8/255, \eta = \varepsilon/4\}$, 2) PGD-200 with $\{k = 200, \varepsilon = 8/255, \eta = \varepsilon/4\}$, 3) Auto-Attack (AA) (Croce & Hein, 2020) with $\varepsilon = 8/255$, which is an ensemble method of four different attacks. The $l_\infty$-norm is used for all measures. Unless otherwise specified, we set $\gamma = 0.1$ to its default value. The ablation study on $\gamma$ is provided in latter part. We repeated all the experiments three times for different seeds. More training details are provided in Appendix A.6.

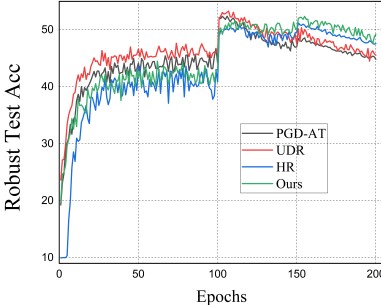
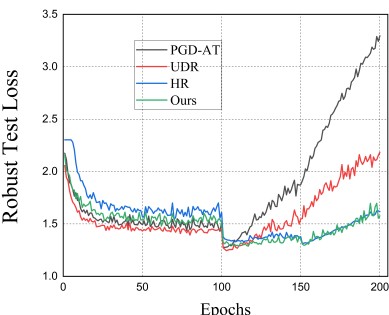

Figure 2: Comparison of SR-WDRO against other robust training methods on CIFAR10 ($\varepsilon = 8/255$). Left: Robust test accuracy. Right: Robust test loss. Our method (green) demonstrates competitive performance in both metrics, particularly in mitigating robust overfitting and higher robust test accuracy.

**Mitigating robust overfitting.** First, we present the adversarial loss and adversarial accuracy on the test set over the 200 training epochs for different methods in Figure 2. It can be observed that the UDR method, similar to standard AT, suffers from severe robust overfitting. This is particularly evident in the adversarial loss on the test set, which increases sharply when robust overfitting occurs. Both HR and our SR-WDRO effectively mitigate this issue, and our method demonstrates superior robust test accuracy. Furthermore, in Table 1, we report the best robust test accuracy and the final

Table 1: Performance of different methods on CIFAR-10 and CIFAR-100 using a ResNet-18 for $l_\infty$ with budget 8/255. We use PGD-10 as the attack to evaluate robust performance. Nat is the natural test accuracy. The 'Best' Robust Test Acc is the highest robust test accuracy achieved during training whereas the 'Final' Robust Test Acc is last epoch's robust accuracy. Best scores are highlighted in boldface.

| Robust Methods | Nat | Robust Test Acc (%) | | |
|---|---|---|---|---|
| | | Final | Best | Diff |
| CIFAR-10 | | | | |
| PGD-AT | $84.80 \pm 0.14$ | $45.16 \pm 0.19$ | $52.91 \pm 0.11$ | $7.75 \pm 0.17$ |
| UDR-AT | $83.87 \pm 0.26$ | $46.60 \pm 0.27$ | $\mathbf{53.23 \pm 0.30}$ | $6.63 \pm 0.57$ |
| HR | $83.95 \pm 0.32$ | $47.32 \pm 0.59$ | $51.23 \pm 0.25$ | $3.90 \pm 0.47$ |
| Ours | $83.34 \pm 0.16$ | $\mathbf{48.58 \pm 0.21}$ | $51.95 \pm 0.19$ | $\mathbf{3.36 \pm 0.06}$ |
| CIFAR-100 | | | | |
| PGD-AT | $\mathbf{57.42 \pm 0.28}$ | $21.87 \pm 0.20$ | $\mathbf{29.37 \pm 0.20}$ | $7.50 \pm 0.35$ |
| UDR-AT | $56.20 \pm 0.54$ | $22.07 \pm 0.12$ | $29.35 \pm 0.02$ | $7.28 \pm 0.10$ |
| HR | $56.69 \pm 0.43$ | $21.15 \pm 0.20$ | $28.35 \pm 0.30$ | $7.20 \pm 0.49$ |
| Ours | $56.71 \pm 0.08$ | $\mathbf{23.09 \pm 0.20}$ | $28.95 \pm 0.29$ | $\mathbf{5.86 \pm 0.50}$ |

robust test accuracy throughout the training process for different methods. The gap between these two metrics is the smallest in our method, indicating that our approach mitigates robust overfitting more effectively. Furthermore, our method remains consistently effective across different attack budgets as shown in Table 4 in the Appendix A.6.

As another popular adversarial defense strategy, TRADES (Zhang et al., 2019) also exhibits overfitting tendencies, albeit to a lesser extent compared to PGD-AT. Our proposed approach effectively mitigates overfitting when applied to UDR-TRADES, as illustrated in Table 5 in Appendix A.6.

**Results on WRN28-10.** We would like to provide further experimental results on the CIFAR-10 dataset with WRN28-10 as shown in Table 2. It can be observed that our method achieves the best performance in mitigating adversarial overfitting, as evidenced by the lowest robustness gap (Diff). Furthermore, it achieves the highest robust accuracy, demonstrating the effectiveness of our approach in enhancing robustness while maintaining strong performance on large models like WideResNet28-10.

Table 2: Robust performance of different robust methods using WideResNet28-10 on CIFAR-10 for $l_\infty$ with budget 8/255.

| Robust Methods | Nat | Robust Test Acc (%) | | |
|---|---|---|---|---|
| | | Final | Best | Diff |
| CIFAR-10 | | | | |
| PGD-AT | $86.51 \pm 0.16$ | $48.81 \pm 0.49$ | $55.64 \pm 0.07$ | $6.83 \pm 0.55$ |
| UDR-AT | $85.99 \pm 0.15$ | $49.01 \pm 0.13$ | $\mathbf{55.82 \pm 0.19}$ | $6.81 \pm 0.31$ |
| HR | $84.89 \pm 0.16$ | $48.45 \pm 0.31$ | $52.45 \pm 0.27$ | $4.00 \pm 0.34$ |
| Ours | $84.52 \pm 0.27$ | $\mathbf{51.26 \pm 0.42}$ | $53.87 \pm 0.06$ | $\mathbf{2.61 \pm 0.47}$ |

**Robustness for smaller $\varepsilon$.** Theorem 5 shows that to ensure robustness on the test set, it is generally necessary to employ a larger attack budget during training compared to testing. In this experiment, we examine the robustness generalization by attacking different robust methods with smaller attack budget than the training attack budget while keeping other parameters of PGD attack the same. The results of this experiment are shown in Table 3. Our method consistently improves robustness for a smaller test adversarial budget. This empirical observation corroborates the findings of Theorem 5, indicating that our approach provides superior generalization guarantees for test set robustness when the training attack budget marginally exceeds that used during testing.

Table 3: Robustness evaluation on CIFAR-10 using ResNet-18 for $l_\infty$ under different attack budget $\varepsilon$. The adversarial training budget is set to be 8/255 consistently. We use stronger attacks such as PGD-200 and Auto-Attack.

| $\varepsilon$ | 8/255 | | 6/255 | | 4/255 | |
|---|---|---|---|---|---|---|
| | PGD-200 | AA | PGD-200 | AA | PGD-200 | AA |
| PGD-AT | $42.86 \pm 0.27$ | $41.62 \pm 0.25$ | $54.14 \pm 0.18$ | $53.30 \pm 0.19$ | $65.76 \pm 0.03$ | $65.18 \pm 0.13$ |
| UDR-AT | $44.59 \pm 0.27$ | $42.81 \pm 0.24$ | $55.29 \pm 0.18$ | $53.80 \pm 0.06$ | $66.04 \pm 0.34$ | $64.92 \pm 0.24$ |
| HR | $45.27 \pm 0.52$ | $41.99 \pm 0.38$ | $56.66 \pm 0.30$ | $53.70 \pm 0.24$ | $67.38 \pm 0.16$ | $65.09 \pm 0.24$ |
| Ours | $\mathbf{46.79 \pm 0.11}$ | $\mathbf{44.06 \pm 0.32}$ | $\mathbf{57.57 \pm 0.53}$ | $\mathbf{55.09 \pm 0.46}$ | $\mathbf{67.56 \pm 0.34}$ | $\mathbf{65.76 \pm 0.31}$ |

**Different choice of $\gamma$.** Figure 3 illustrates the effect of varying values of statistical error $\gamma$ in our framework on mitigating robust overfitting. We observe that larger $\gamma$ values demonstrate reduced overfitting tendencies, particularly evident in the loss curves. However, excessively large $\gamma$ leads to a decrease in natural test accuracy, which drops from $84.8\%(\gamma = 0)$ to $80.4\%(\gamma = 0.2)$. Consequently, we default to $\gamma = 0.1$ as a trade-off between robust overfitting mitigation and natural test accuracy preservation.

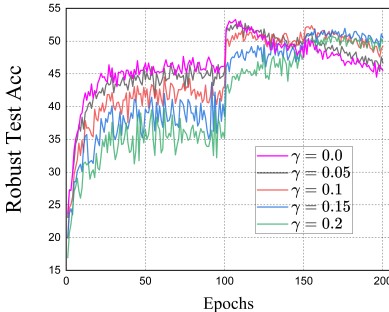 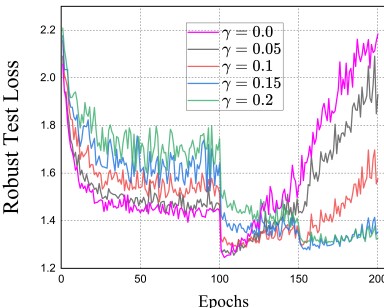

Figure 3: Impact of statistical error $\gamma$ on mitigating overfitting in our method. Experiments conducted on CIFAR10 with $\varepsilon = 8/255$. Left: Robust test accuracy. Right: Robust test loss.

## 7 CONCLUSIONS

In this paper, we introduce SR-WDRO, a novel approach to address the challenge of robust overfitting in WDRO. Our method combines Kullback-Leibler divergence and Wasserstein distance to create a new ambiguity set, providing theoretical guarantees that adversarial test loss can be upper bounded by the statistically robust training loss with high probability and establishing conditions for Stackelberg and Nash equilibria between the learner and adversary. We developed a practical training algorithm based on this framework, which maintains computational efficiency comparable to standard adversarial training methods. Extensive experiments on benchmark datasets demonstrated that SR-WDRO effectively mitigates robust overfitting and achieves superior adversarial robustness compared to existing approaches. Our work contributes both theoretical insights and practical advancements to the field of robust machine learning. The proposed method offers a promising direction for developing more reliable and robust models in the face of distributional shifts.

**Limitations and future works.** Our SR-WDRO mitigates robust overfitting and improves robust accuracy compared to existing methods, although compromises with some natural accuracy drop and slight computational overhead. Due to the intractability of Equations (4) and (8), a better approximation is welcomed to solve SR-WDRO to mitigate the compromise of accuracy and computational cost. Furthermore, we mainly focus on supervised learning in this paper, expanding SR-WDRO to broader tasks including unsupervised learning, regressive tasks is a promising direction.

ACKNOWLEDGMENTS

This work is supported by NSFC grant No.12288201 and No.92270001, and grant GJ0090202. The authors thank anonymous referees for their valuable comments.

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

## A APPENDIX

### A.1 RELATIONSHIP BETWEEN DIFFERENT DISCREPANCIES

We first introduce Levy-Prokhorov and total variation metrics. Then we will establish relationships among the four metrics: the Wasserstein metric, Levy-Prokhorov metric, total variation metric, and KL divergence.

**Definition 13.** *The Levy-Prokhorov metric between $\mu, \nu \in \mathcal{P}(\mathcal{Z})$ is:*

$$\mathrm{LP}(\mu, \nu) := \inf \{\tau > 0 \mid \mu(A) \le \nu(A^\tau) + \tau \quad \forall A \in \mathcal{B}(\mathcal{Z})\}$$

*where $A^\tau = \{z \in \mathcal{Z} : \inf_{z' \in A} \mathrm{d}(z, z') \le \tau\}$.*

**Definition 14.** *The total variation distance between any $\mu, \nu \in \mathcal{P}(\mathcal{Z})$ is:*

$$\mathrm{TV}(\mu, \nu) = \inf \{\tau > 0 \mid \mu(A) \le \nu(A) + \tau \quad \forall A \in \mathcal{B}(\mathcal{Z})\}.$$

The KL divergence (relative entropy) has some basic properties: (1) $\mathrm{KL}(\mu, \nu) \ge 0$, which equal to 0 if and only if $\mu = \nu$; (2) KL is not symmetric and does not satisfy the triangle inequality.

Here, we give some relationships between these discrepancies.

**Proposition 15.** *For any $\mu, \nu \in \mathcal{P}(\mathcal{Z})$, the Wasserstein metric and Levy-Prokhorov metric satisfy:*

$$\mathrm{LP}(\mu, \nu)^{\frac{p+1}{p}} \le \mathrm{W}_p(\mu, \nu) \le (\mathrm{diam}(\mathcal{Z}) + 1)\mathrm{LP}(\mu, \nu)^{\frac{1}{p}}.$$

*Proof.* For any joint distribution $\pi$ on random variables $X, Y$,

$$\mathbb{E}_\pi[\mathrm{d}^p(X, Y)] \le \tau^p \cdot \mathbb{P}(\mathrm{d}(X, Y) \le \tau) + \mathrm{diam}(\mathcal{Z})^p \cdot \mathbb{P}(\mathrm{d}(X, Y) > \tau)$$
$$\le \tau^p + (\mathrm{diam}(\mathcal{Z})^p - \tau^p) \cdot \mathbb{P}(\mathrm{d}(X, Y) > \tau).$$

If $\mathrm{LP}(\mu, \nu) \le \tau (\tau \in [0, 1])$, we can choose a coupling of $\mu$ and $\nu$ so that $\mathbb{P}(\mathrm{d}(X, Y) > \tau)$ is upper bounded by $\tau$ (Huber, 1981). Then we have:

$$\mathbb{E}_\pi[\mathrm{d}^p(X, Y)] \le \tau^p + (\mathrm{diam}(\mathcal{Z})^p - \tau^p)\tau \le \tau(\mathrm{diam}(\mathcal{Z})^p + 1).$$

Take the infimum of both sides over coupling, we have:

$$\mathrm{W}_p(\mu, \nu) = \inf_\pi (\mathbb{E}_\pi(\mathrm{d}^p(X, Y))^{\frac{1}{p}} \le (\tau(\mathrm{diam}(\mathcal{Z})^p + 1))^{\frac{1}{p}} \le \tau^{\frac{1}{p}}(\mathrm{diam}(\mathcal{Z}) + 1).$$

In order to bound Levy-Prokhorov metric by Wasserstein metric, choose $\tau$ such that $\mathrm{W}_p(\mu, \nu) = \tau^{\frac{p+1}{p}}$, and use Markov's inequality. We have

$$\mathbb{P}(\mathrm{d}(X, Y) > \tau) = \mathbb{P}(\mathrm{d}^p(X, Y) > \tau^p) \le \frac{1}{\tau^p}\mathbb{E}_\pi[\mathrm{d}^p(X, Y)] \le \tau$$

where $\pi$ is any joint distribution on $X \times Y$. Then by Strassen's theorem (Huber, 1981, Theorem 3.7), $\mathbb{P}(\mathrm{d}(X, Y) > \tau) \le \tau$ is equivalent to $\mu(A) \le \nu(A^\tau) + \tau$ for all Borel set $A \in \mathbb{B}(\mathcal{Z})$, which means $\mathrm{LP}(\mu, \nu) \le \tau$, thus

$$\mathrm{LP}(\mu, \nu)^{\frac{p+1}{p}} \le \mathrm{W}_p(\mu, \nu).$$

$\square$

**Proposition 16.** *For any $\mu, \nu \in \mathcal{P}(\mathcal{Z})$, the Wasserstein metric, total variation metric and KL divergence satisfy:*

$$\mathrm{W}_p(\mu, \nu) \le \mathrm{diam}(\mathcal{Z}) \cdot \mathrm{TV}(\mu, \nu)^{\frac{1}{p}} \le \mathrm{diam}(\mathcal{Z}) \cdot \left(\frac{\mathrm{KL}(\mu, \nu)}{2}\right)^{\frac{1}{2p}}.$$

*Proof.* The total variation distance is equivalent to $W_1(\mu, \nu)$ with the optimal transport cost $\mathrm{d}(z, z') = \mathbb{I}(z \ne z')$ by definition, where $\mathbb{I}$ is the indicator function. As $\mathrm{d}(z, z') \le \mathrm{diam}(\mathcal{Z}) \cdot \mathbb{I}(z \ne$

$z'$), we have:

$$
\begin{aligned}
\mathrm{W}_p(\mu, \nu) &= \inf_{\pi \in \Pi(\mu, \nu)} \left\{ \mathbb{E}_{(z, z') \sim \pi} \left[ \mathrm{d}^p(z, z') \right] \right\}^{\frac{1}{p}} \\
&\leq \inf_{\pi \in \Pi(\mu, \nu)} \left\{ \mathbb{E}_{(z, z') \sim \pi} \left[ \mathrm{diam}(\mathcal{Z})^p \cdot \mathbb{I}(z \neq z') \right] \right\}^{\frac{1}{p}} \\
&= \mathrm{diam}(\mathcal{Z}) \inf_{\pi \in \Pi\mu, \nu)} \left\{ \mathbb{E}_{(z, z') \sim \pi} \left[ \mathbb{I}(z, z') \right] \right\}^{\frac{1}{p}} \\
&= \mathrm{diam}(\mathcal{Z}) \cdot \mathrm{TV}(\mu, \nu)^{\frac{1}{p}}.
\end{aligned}
$$

And the second part is by Pinsker's inequality that $\mathrm{TV}(\mu, \nu) \leq \sqrt{\frac{1}{2} \mathrm{KL}(\mu, \nu)}$. $\qquad \square$

### A.2 PROOFS OF THEOREMS 3 AND 5

In this subsection, we prove formally Theorem 3 and Theorem 5. The goal is to show that our SR-WDRO training loss is an upper bound on the test loss with high probability.

**Lemma 17.** *Let $\mathcal{D}_n$ be the observed empirical distribution of $n$ independent samples with true distribution $\mathcal{D}$ on a compact space $\mathcal{Z}$. Then for all $\varepsilon > 0$, let $\delta = (\frac{\varepsilon}{\mathrm{diam}(\mathcal{Z})+1})^p$. We have*

$$
\mathbb{P}\left( \exists \mathcal{D}' \in \mathcal{P}(\mathcal{Z}), \ \mathrm{W}_p(\mathcal{D}_n, \mathcal{D}') \leq \varepsilon, \ \mathrm{KL}(\mathcal{D}', \mathcal{D}) \leq \gamma \right) \geq 1 - e^{-\gamma n} \left( \frac{4}{\delta} \right)^{m(\mathcal{Z}, \delta)}
$$

*where $m(\mathcal{Z}, \delta) := \min\{ k \geq 0 : \exists \xi_1, \cdots, \xi_k \in \mathcal{Z}, \ s.t. \ \cup_{i=1}^k \mathbb{B}(\xi_i, \delta) \supseteq \mathcal{Z} \}$ denote the internal covering number of the support set $\mathcal{Z}$.*

*Proof.* We have:

$$
\begin{aligned}
&\mathbb{P}\left( \exists D' \in \mathcal{P}(\mathcal{Z}), \ \mathrm{W}_p\left( \mathcal{D}_n, \mathcal{D}' \right) \leq \varepsilon, \ \mathrm{KL}\left( \mathcal{D}', \mathcal{D} \right) \leq \gamma \right) \\
&= \mathbb{P}\left( \mathcal{D}_n \in \left\{ \hat{\mathcal{D}} \in \mathcal{P}(\mathcal{Z}) : \exists \mathcal{D}' \in \mathcal{P}(\mathcal{Z}) \text{ s.t. } \mathrm{W}_p(\hat{\mathcal{D}}, \mathcal{D}') \leq \varepsilon, \ \mathrm{KL}(\mathcal{D}', \mathcal{D}) \leq \gamma \right\} \right) \\
&= 1 - \mathbb{P}\left( \mathcal{D}_n \in \mathcal{A} \right)
\end{aligned}
$$

where $\mathcal{A}$ is defined as: $\mathcal{A}^c = \left\{ \hat{\mathcal{D}} \in \mathcal{P}(\mathcal{Z}) : \exists \mathcal{D}' \in \mathcal{P}(\mathcal{Z}) \text{ s.t. } \mathrm{W}_p(\hat{\mathcal{D}}, \mathcal{D}') \leq \varepsilon, \ \mathrm{KL}\left( \mathcal{D}', \mathcal{D} \right) \leq \gamma \right\}.$

Let $\mathcal{A}^\delta = \{ \mathcal{D}' \in \mathcal{P}(\mathcal{Z}) : \mathrm{LP}\left( D', D'' \right) \leq \delta, D'' \in \mathcal{A} \}$ is the $\delta$-inflation of the set $\mathcal{A}$ with respect to the LP metric. Denote $\mathbb{B}_{\mathrm{LP}}\left( \mathcal{D}' \right) = \{ D'' \in \mathcal{P}(\mathcal{Z}) : \mathrm{LP}\left( D', D'' \right) \leq \delta \}$ the LP ball, which is compact as LP is continuous in the weak topology and $\mathcal{Z}$ is compact (Prokhorov, 1956).

Dembo (2009) have established that for any set $\mathcal{A} \subset \mathcal{P}(\mathcal{Z})$ and $\delta > 0$ we have for all $n \geq 1$ the upper bound:

$$
\mathbb{P}\left( \mathcal{D}_n \in \mathcal{A} \right) \leq m_{\mathrm{LP}}(\mathcal{A}, \delta) \exp\left( -n \inf_{\mathcal{D}' \in \mathcal{A}^\delta} \mathrm{KL}\left( \mathcal{D}', \mathcal{D} \right) \right)
$$

where $m_{\mathrm{LP}}(\mathcal{A}, \delta) = \min\{ k \geq 0 : \exists \mathcal{D}_1 \cdots \mathcal{D}_k \in \mathcal{A} \text{ s.t. } \bigsqcup_{i=1}^k \mathbb{B}_{\mathrm{LP}}\left( \mathcal{D}_i, \delta \right) \supset \mathcal{A} \}$ is the internal covering number of set $\mathcal{A}$ with LP balls of radius $\delta$. Furthermore, we can upper bound the internal covering number of any set $\mathcal{A}$ in terms of the internal covering number of the compact event $\mathcal{Z}$ as $m_{\mathrm{LP}}(\mathcal{A}, \delta) \leq m_{\mathrm{LP}}(\mathcal{P}(\mathcal{Z}), \delta) \leq (4/\delta)^{m(\mathcal{Z}, \delta)}$ for all $\delta > 0$ (Dembo, 2009). Hence, we have:

$$
\mathbb{P}\left( \mathcal{D}_n \in \mathcal{A} \right) \leq (4/\delta)^{m(\mathcal{Z}, \delta)} \cdot \exp\left( -n \min_{\mathcal{D}' \in \mathcal{A}^\delta} \mathrm{KL}\left( \mathcal{D}', \mathcal{D} \right) \right).
$$

The final inequality immediately leads to conclusion by remarking that $\mathcal{D}' \in \mathcal{A}^\delta \Rightarrow \mathrm{KL}\left( \mathcal{D}', \mathcal{D} \right) > \gamma$. Otherwise, suppose that there exists $\mathcal{D}'' \in \mathcal{A}^\delta$ with $\mathrm{KL}\left( \mathcal{D}'', \mathcal{D} \right) \leq \gamma$. By definition of $\mathcal{A}^\delta$, there exists $\mathcal{D}' \in \mathcal{A}$ such that $\mathrm{LP}(\mathcal{D}'', \mathcal{D}') = \mathrm{LP}(\mathcal{D}', \mathcal{D}'') \leq \delta$. And by Proposition 15, we have $\mathrm{W}_p\left( \mathcal{D}', \mathcal{D}'' \right) \leq (\mathrm{diam}(\mathcal{Z}) + 1)\mathrm{LP}(\mathcal{D}', \mathcal{D}'')^{\frac{1}{p}} = \varepsilon$, so we have both $\mathrm{W}_p\left( \mathcal{D}', \mathcal{D}'' \right) \leq \varepsilon$ and $\mathrm{KL}\left( \mathcal{D}'', \mathcal{D} \right) \leq \gamma$, which implies that $\mathcal{D}' \in \mathcal{A}^c$, it is a contradiction. $\qquad \square$

**Proof of Theorem 3** We restate the theorem below. Let $\mathcal{D}_n$ be the observed empirical distribution which may be corrupted, and $\mathcal{D}$ be the true data distribution. Then for all $\varepsilon > 0$, let $\delta = (\frac{\varepsilon}{\text{diam}(\mathcal{Z})+1})^p$, we have

$$\mathbb{P}\Big(\forall \theta \in \Theta, \mathcal{L}_{\varepsilon,\gamma}(\theta) \geq \mathbb{E}_{\mathcal{D}}[L(\theta, z)]\Big) \geq 1 - e^{-\gamma n} \left(\frac{4}{\delta}\right)^{m(\mathcal{Z},\delta)} \tag{10}$$

where $m(\mathcal{Z}, \delta) := \min\{k \geq 0 : \exists \xi_1, \cdots, \xi_k \in \mathcal{Z}, \text{ s.t. } \cup_{i=1}^{k} \mathbb{B}(\xi_i, \delta) \supseteq \mathcal{Z}\}$ denote the internal covering number of the support set $\mathcal{Z}$.

*Proof.* Recall that the ambiguity set is $\mathcal{U}(\mathcal{D}_n) := \{\mathcal{D}' : \exists \mathcal{D}'' \in \mathcal{P}(\mathcal{Z}) \text{ s.t. } W_p(\mathcal{D}_n, \mathcal{D}'') \leq \varepsilon, \text{ KL}(\mathcal{D}'', \mathcal{D}') \leq \gamma\}$. By Lemma 17 we can ensure the ambiguity set $\mathcal{U}(\mathcal{D}_n)$ contains the true distribution $\mathcal{D}$ with high probability:

$$\mathbb{P}\left(\mathcal{D} \in \mathcal{U}(\mathcal{D}_n)\right) \geq 1 - e^{-\gamma n} \left(\frac{4}{\delta}\right)^{m(\mathcal{Z},\delta)}.$$

It follows that

$$\mathbb{P}\left(\sup_{\mathcal{D}' \in \mathcal{U}(\mathcal{D}_n)} \mathbb{E}_{\mathcal{D}'}[L(\theta, z)] \geq \mathbb{E}_{\mathcal{D}}[L(\theta, z)], \forall \theta \in \Theta\right) \geq 1 - e^{-\gamma n} \left(\frac{4}{\delta}\right)^{m(\mathcal{Z},\delta)}.$$

$\square$

To facilitate our analysis on test adversarial distribution, we introduce the Levy-Prokhorov (LP) metric given by Bennouna et al. (2023), $\text{LP}_\varepsilon$ is defined as:

$$\text{LP}_\varepsilon(\mathcal{D}, \mathcal{D}') := \inf\left\{\int \mathbf{1}\left(d(z, z') > \varepsilon\right) d\pi(z, z') : \pi \in \Pi(\mathcal{D}, \mathcal{D}')\right\}. \tag{11}$$

**Proof of the Theorem 5** We restate the theorem below. Let $\mathcal{D}$ be the true data distribution, $\mathcal{D}_n$ the empirical distribution sampled i.i.d. from $\mathcal{D}$, and $\delta = (\frac{\sigma}{\text{diam}(\mathcal{Z})+1})^p$ for any $\sigma > 0$. Then

$$\mathbb{P}\Big(\forall \theta \in \Theta, \mathbb{E}_{\mathcal{D}}[\max_{z' \in \mathbb{B}(z;\epsilon)} L(\theta, z')] \leq \mathcal{L}_{\varepsilon+\sigma,\gamma}(\theta, \mathcal{D}_n)\Big) \geq 1 - e^{-n\gamma} \left(\frac{4}{\delta}\right)^{m(\mathcal{Z},\delta)} \tag{12}$$

where $m(\mathcal{Z}, \delta) := \min\{k \geq 0 : \exists \xi_1, \cdots, \xi_k \in \mathcal{Z}, \text{ s.t. } \cup_{i=1}^{k} \mathbb{B}(\xi_i, \delta) \supseteq \mathcal{Z}\}$ denote the internal covering number of the support set $\mathcal{Z}$.

*Proof.* We consider the adversarial test distribution $\mathcal{M}_\# \mathcal{D}$, where $\mathcal{M}$ maps every input $z$ to $\arg\max_{z' \in \mathbb{B}(z;\epsilon)} L(\theta, z')$. It is clear that $\mathcal{M}_\# \mathcal{D} \in \{\mathcal{D}' \in \mathcal{P}(\mathcal{Z}) : \text{LP}_\varepsilon(\mathcal{D}, \mathcal{D}') \leq 0\}$. According to Lemma 17, we have that for $\delta = (\frac{\sigma}{\text{diam}(\mathcal{Z})+1})^p$:

$$\mathbb{P}\Big(\exists \mathcal{D}' \in \mathcal{P}(\mathcal{Z}), W_p(\mathcal{D}_n, \mathcal{D}') \leq \sigma, \text{ KL}(\mathcal{D}', \mathcal{D}) \leq \gamma\Big) \geq 1 - e^{-\gamma n} \left(\frac{4}{\delta}\right)^{m(\mathcal{Z},\delta)}.$$

Let us denote here $\mathcal{U}'_{\sigma,\gamma,\varepsilon}(\mathcal{D}_n) := \{\mathcal{D}'_{\text{test}} : \exists \mathcal{D}', \mathcal{D}'' \in \mathcal{P}(\mathcal{Z}) \text{ s.t. } W_p(\mathcal{D}_n, \mathcal{D}') \leq \sigma, \text{ KL}(\mathcal{D}', \mathcal{D}'') \leq \gamma, \text{ LP}_\varepsilon(\mathcal{D}'', \mathcal{D}'_{\text{test}}) \leq 0\}$, by the definition of $\mathcal{M}_\# \mathcal{D}$, we have

$$\mathbb{P}\left(\mathcal{M}_\# \mathcal{D} \in \mathcal{U}'_{\sigma,r,\varepsilon}(\mathcal{D}_n)\right) \geq 1 - e^{-\gamma n} \left(\frac{4}{\delta}\right)^{m(\mathcal{Z},\delta)}.$$

It follows immediately that:

$$\mathbb{P}\left(\sup_{\mathcal{D}'_{\text{test}} \in \mathcal{U}'_{\sigma,\gamma,\varepsilon}(\mathcal{D}_n)} \mathbb{E}_{\mathcal{D}'_{\text{test}}}[L(\theta, z)] \geq \mathbb{E}_{\mathcal{M}_\# \mathcal{D}}[L(\theta, z)], \forall \theta \in \Theta\right) \geq 1 - e^{-\gamma n} \left(\frac{4}{\delta}\right)^{m(\mathcal{Z},\delta)}.$$

And we have

$$\mathbb{E}_{\mathcal{M}_\# \mathcal{D}} L(\theta, z) = \mathbb{E}_{\mathcal{D}}[\max_{z' \in \mathbb{B}(z;\epsilon)} L(\theta, z')].$$

It remains to show that $\sup_{\mathcal{D}'_{\text{test}} \in \mathcal{U}'_{\sigma,\gamma,\varepsilon}(\mathcal{D}_n)} \mathbb{E}_{\mathcal{D}'_{\text{test}}}[L(\theta,z)] \leq \mathcal{L}_{\varepsilon+\sigma,\gamma}(\theta,\mathcal{D}_n)$ for all $\theta$. We have

$$\sup_{\mathcal{D}'_{\text{test}} \in \mathcal{U}'_{\sigma,\gamma,\varepsilon}(\mathcal{D}_n)} \mathbb{E}_{\mathcal{D}'_{\text{test}}}[L(\theta,z)]$$

$$= \sup_{\mathcal{D}':\text{W}_p(\mathcal{D}_n,\mathcal{D}') \leq \sigma} \sup\{\mathbb{E}_{\mathcal{D}'_{\text{test}}}[L(\theta,z)] : \exists \mathcal{D}'', \mathcal{D}'_{\text{test}} \text{ s.t. } \text{KL}(\mathcal{D}',\mathcal{D}'') \leq \gamma, \ \text{LP}_\varepsilon(\mathcal{D}'',\mathcal{D}'_{\text{test}}) \leq 0\}$$

$$\overset{(1)}{=} \sup_{\mathcal{D}':\text{W}_p(\mathcal{D}_n,\mathcal{D}') \leq \sigma} \sup\{\mathbb{E}_{\mathcal{D}'_{\text{test}}}[L(\theta,z)] : \exists \mathcal{D}'', \mathcal{D}'_{\text{test}} \text{ s.t. } \text{LP}_\varepsilon(\mathcal{D}',\mathcal{D}'') \leq 0, \ \text{KL}(\mathcal{D}''; ,\mathcal{D}'_{\text{test}}) \leq \gamma\}$$

$$= \sup\{\mathbb{E}_{\mathcal{D}'_{\text{test}}}[L(\theta,z)] : \exists \mathcal{D}', \mathcal{D}'' \text{ s.t. } \text{W}_p(\mathcal{D}_n,\mathcal{D}') \leq \sigma, \text{LP}_\varepsilon(\mathcal{D}',\mathcal{D}'') \leq 0, \ \text{KL}(\mathcal{D}''; ,\mathcal{D}'_{\text{test}}) \leq \gamma\}$$

$$\overset{(2)}{\leq} \sup\{\mathbb{E}_{\mathcal{D}'_{\text{test}}}[L(\theta,z)] : \exists \mathcal{D}', \mathcal{D}'' \text{ s.t. } \text{W}_p(\mathcal{D}_n,\mathcal{D}') \leq \sigma, \text{W}_p(\mathcal{D}',\mathcal{D}'') \leq \varepsilon, \ \text{KL}(\mathcal{D}''; ,\mathcal{D}'_{\text{test}}) \leq \gamma\}$$

$$\overset{(3)}{\leq} \sup\{\mathbb{E}_{\mathcal{D}'_{\text{test}}}[L(\theta,z)] : \exists \mathcal{D}' \text{ s.t. } \text{W}_p(\mathcal{D}_n,\mathcal{D}') \leq \varepsilon+\sigma, \ \text{KL}(\mathcal{D}',\mathcal{D}'_{\text{test}}) \leq \gamma\}$$

$$\leq \mathcal{L}_{\varepsilon+\sigma,\gamma}(\theta,\mathcal{D}_n).$$

The first equality follows from Lemma C.3 in Bennouna et al. (2023), the second inequality is clear as under $\text{LP}_\varepsilon$ constraint we have $\text{d}(z,z') \leq \varepsilon$ for any $z,z'$ almost surely. The third inequality follows from the triangle inequality. □

### A.3 GENERAL DISTRIBUTIONAL SHIFTS

Our framework is applicable to a variety of distributional shifts, not limited to the adversarial example distributions that we focus on. This includes other shifts such as those encountered in domain generalization. If the distance between the test distribution and the original data distribution remains within certain bounds, we can still provide generalization guarantees. Here we assume that

$$\mathcal{D}_{\text{test}} \in \{\mathcal{D}' \in \mathcal{P}(\mathcal{Z}) : \text{LP}_\varepsilon(\mathcal{D},\mathcal{D}') \leq 0\},$$

where $\varepsilon > 0$ is the adversarial budget and $\text{LP}_\varepsilon$ is defined in equation 11. Here, we consider a more general scenario where potential shifts might include corruptions or other variations, resulting in discrepancies between test samples and true samples. This formulation allows us to address a broader range of distributional shifts beyond just adversarial examples. Similar to Theorem 5, we have:

**Theorem 18.** *Let $\mathcal{D}$ be the true data distribution, $\mathcal{D}_n$ the empirical distribution sampled i.i.d. from $\mathcal{D}$, $\mathcal{D}_{\text{test}}$ the test distribution as defined above, and $\delta = (\frac{\sigma}{\text{diam}(\mathcal{Z})+1})^p$ for $\sigma > 0$. Then*

$$\mathbb{P}\Big(\forall \theta \in \Theta, \mathbb{E}_{\mathcal{D}_{\text{test}}}[L(\theta,z)] \leq \mathcal{L}_{\varepsilon+\sigma,\gamma}(\theta,\mathcal{D}_n)\Big) \geq 1 - e^{-n\gamma}\left(\frac{4}{\delta}\right)^{m(\mathcal{Z},\delta)} \tag{13}$$

*where $m(\mathcal{Z},\delta) := \min\{k \geq 0 : \exists \xi_1, \cdots, \xi_k \in \mathcal{Z}, \text{ s.t. } \cup_{i=1}^k \mathbb{B}(\xi_i,\delta) \supseteq \mathcal{Z}\}$ denote the internal covering number of the support set $\mathcal{Z}$.*

### A.4 PROOFS FOR SECTION 4.3

In this subsection, we first give the definition of Nash equilibrium and Stackelberg equilibrium, then we provide the detailed proof of the existence of two game equilibrium.

Here we consider two player zero sum game, let the strategy spaces of player 1 and player 2 be $X$ and $Y$, respectively. Their utility functions are denoted by $u_1(x,y)$ and $u_2(x,y)$, where $x \in X$ is the strategy of player 1, and $y \in Y$ is the strategy of player 2. In a zero-sum game, $u_1(x,y) = -u_2(x,y)$.

**Definition 19** (Nash Equilibrium). *A Nash equilibrium is a pair of strategies $(x^*, y^*)$ such that neither player can improve their payoff by unilaterally changing their strategy, given the strategy of the other player.*

$$u_1(x^*,y^*) \geq u_1(x,y^*), \quad \forall x \in X; \quad u_1(x^*,y^*) \leq u_1(x^*,y), \quad \forall y \in Y.$$

In a Stackelberg game, one player (the leader) moves first, and the other player (the follower) observes the leader's action before selecting their own strategy. Let player 1 be the leader and player 2 be the follower.

**Definition 20** (Stackelberg Equilibrium). *A Stackelberg equilibrium is a pair of strategies $(x^*, y^*(x^*))$, where $x^*$ is the leader's optimal strategy and $y^*(x^*)$ is the follower's best response given the leader's strategy.*

*1. Follower's best response: $y^*(x) = \arg\max_{y \in Y} u_2(x, y)$*

*2. Leader's optimal strategy: $x^* = \arg\max_{x \in X} u_1(x, y^*(x))$.*

Thus, the Stackelberg equilibrium consists of the leader's strategy $x^*$, which maximizes the leader's payoff, and the follower's response $y^*(x^*)$, which is the best response to $x^*$.

**Proof of Proposition 9:** (Restate) The ambiguity set $\mathcal{U}(\mathcal{D}_n) := \{\mathcal{D}' \in \mathcal{P}(\mathcal{Z}) : \exists \mathcal{D}'' \in \mathcal{P}(\mathcal{Z}) \text{ s.t. } W_p(\mathcal{D}_n, \mathcal{D}'') \leq \varepsilon, \text{ KL}(\mathcal{D}'', \mathcal{D}') \leq \gamma\}$ is compact in $\mathcal{P}(\mathcal{Z})$ with respect to weak topology. And if the loss function satisfies Assumption 8, then the supremum in (4) is finite and can be attained for any $\theta \in \Theta$.

*Proof.* Firstly, we prove $\mathcal{U}(\mathcal{D}_n)$ is closed. Assume that any sequence $\{\mathcal{D}^i\}_{i=1}^{\infty} \subset \mathcal{U}(\mathcal{D}_n)$ weakly converges to $\mathcal{D}^0$. We will show that $\mathcal{D}^0 \in \mathcal{U}(\mathcal{D}_n)$, which is equivalent to $\mathcal{U}(\mathcal{D}_n)$ is closed. By definition of $\mathcal{U}(\mathcal{D}_n)$, for each $i$, there exists $\widehat{\mathcal{D}^i}$ such that $W_p(\mathcal{D}_n, \widehat{\mathcal{D}^i}) \leq \varepsilon$, $\text{KL}(\widehat{\mathcal{D}^i}, \mathcal{D}^i) \leq \gamma$. We aim to show that such $\widehat{\mathcal{D}^0}$ also exist for the limit $\mathcal{D}^0$, preserving the Wasserstein and KL constraints.

The set $\{\mathcal{D}'' \in \mathcal{P}(\mathcal{Z}) : W_p(\mathcal{D}_n, \mathcal{D}'') \leq \varepsilon\}$ is compact in the weak topology of $\mathcal{P}(\mathcal{Z})$, as the Wasserstein ball is compact with respect to weak convergence. Therefore, from the sequence $\widehat{\mathcal{D}^i}$, we can extract a subsequence $\{\widehat{\mathcal{D}^{i_j}}\}_{j=1}^{\infty}$ that converges weakly to some distribution $\widehat{\mathcal{D}^0}$, and by the compactness of the Wasserstein ball, $\widehat{\mathcal{D}^0}$ satisfies: $W_p(\mathcal{D}_n, \widehat{\mathcal{D}^0}) \leq \varepsilon$. Here, we abuse the notation and still denote the subsequence as $\{\widehat{\mathcal{D}^i}\}_{i=1}^{\infty}$.

Next, we use the lower semicontinuity property of the Kullback-Leibler (KL) divergence with respect to weak convergence. Since $\text{KL}(\widehat{\mathcal{D}^i}, \mathcal{D}^i) \leq \gamma$ for all $i$, and $\widehat{\mathcal{D}^i} \to \widehat{\mathcal{D}^0}$ weakly and $\mathcal{D}^i \to \mathcal{D}^0$ weakly, we apply the lower semicontinuity of the KL divergence to conclude:
$$\text{KL}(\widehat{\mathcal{D}^0}, \mathcal{D}^0) \leq \liminf_{i \to \infty} \text{KL}(\widehat{\mathcal{D}^i}, \mathcal{D}^i) \leq \gamma.$$

Thus, the limit point $\mathcal{D}^0$ of the sequence $\{\mathcal{D}^i\}_{i=1}^{\infty}$ satisfies the conditions:
$$\exists \widehat{\mathcal{D}^0} \in \mathcal{P}(\mathcal{Z}) \text{ s.t. } W_p(\mathcal{D}_n, \widehat{\mathcal{D}^0}) \leq \varepsilon \quad \text{and} \quad \text{KL}(\widehat{\mathcal{D}^0}, \mathcal{D}^0) \leq \gamma,$$

which implies that $\mathcal{D}^0 \in \mathcal{U}(\mathcal{D}_n)$. Therefore, $\mathcal{U}(\mathcal{D}_n)$ is closed with respect to the weak topology.

Then we prove that $\mathcal{U}(\mathcal{D}_n)$ is subset of a larger Wasserstein ball, as any weak closed subset of a weakly compact set is weakly compact, we can prove $\mathcal{U}(\mathcal{D}_n)$ is weak compact. For any $\mathcal{D}' \in \mathcal{U}(\mathcal{D}_n)$, there exists $\mathcal{D}'' \in \mathcal{P}(\mathcal{Z})$ such that $W_p(\mathcal{D}_n, \mathcal{D}'') \leq \varepsilon$ and $\text{KL}(\mathcal{D}'', \mathcal{D}') \leq \gamma$. By Proposition 16, we have $W_p(\mathcal{D}'', \mathcal{D}') \leq \text{diam}(\mathcal{Z})(\gamma/2)^{\frac{1}{2p}}$, then we have that $W_p(\mathcal{D}_n, \mathcal{D}'') \leq \varepsilon$ and $W_p(\mathcal{D}'', \mathcal{D}') \leq \text{diam}(\mathcal{Z})(\gamma/2)^{\frac{1}{2p}}$, thus $W_p(\mathcal{D}_n, \mathcal{D}') \leq \varepsilon + \text{diam}(\mathcal{Z})(\gamma/2)^{\frac{1}{2p}}$, which means that $\mathcal{U}(\mathcal{D}_n) \subset \mathbb{B}_{W_p}\left(\mathcal{D}_n, \ \varepsilon + \gamma \cdot \text{diam}(\mathcal{Z})(\gamma/2)^{\frac{1}{2p}}\right)$.

Next, we show the supremum in (4) is finite. As $\mathcal{U}(\mathcal{D}_n) \subset \mathbb{B}_{W_p}(\mathcal{D}_n, \ \varepsilon + \gamma a \cdot d_{\min}^p(\mathcal{Z}))$, it is clear that:
$$\sup_{\mathcal{D}' \in \mathcal{U}(\mathcal{D}_n)} \mathbb{E}_{\mathcal{D}'}[L(\theta, z)] \leq \sup_{\mathcal{D}' \in \mathbb{B}_{W_p}(\mathcal{D}_n, \ \varepsilon + \gamma a \cdot d_{\min}^p(\mathcal{Z}))} \mathbb{E}_{\mathcal{D}'}[L(\theta, z)].$$
The supremum on the right side of the above inequality is finite by Yue et al. (2022, Theorem 2), which applies Assumption 8 (iii) and $\mathcal{D}_n$ has a finite $p$-th moment. Thus the supremum have a finite upper bound.

It remains to be shown that supremum in (4) can be attained. We know that the objective function $\sup_{\mathcal{D}' \in \mathcal{U}(\mathcal{D}_n)} \mathbb{E}_{\mathcal{D}'}[L(\theta, z)]$ is weak upper semi-continuous in $\mathcal{D}'$ over the ambiguity set $\mathcal{U}(\mathcal{D}_n)$. This follows immediately from the proof of Yue et al. (2022, Theorem 3), which reveals that $\sup_{\mathcal{D}' \in \mathcal{U}(\mathcal{D}_n)} \mathbb{E}_{\mathcal{D}'}[L(\theta, z)]$ is weak upper semi-continuous over $\mathbb{B}_{W_p}(\mathcal{D}_n, \ \varepsilon + \gamma a \cdot d_{\min}^p(\mathcal{Z}))$. As $\mathcal{U}(\mathcal{D}_n)$ is weak compact, Weirestrass's theorem the guarantees the supremum in (4) is indeed attained for any $\theta$. □

**Proof of the Stackelberg Equilibrium (Theorem 10)**  (Restate) If the loss function satisfies Assumption 8 and we assume $\Theta$ is compact, the game exists a Stackelberg equilibrium. i.e. there exists

$$\theta^* \in \arg\inf_{\theta \in \Theta} \sup_{\mathcal{D}' \in \mathcal{U}(\mathcal{D}_n)} \mathbb{E}_{\mathcal{D}'}[L(\theta, z)], \quad \mathcal{D}'^*(\theta^*) \in \arg\sup_{\mathcal{D}' \in \mathcal{U}(\mathcal{D}_n)} \mathbb{E}_{\mathcal{D}'}[L(\theta^*, z)].$$

*Proof.* By Proposition 9, for any $\theta$, the supremum in 4 can be attained, that is, there exists a best response $\mathcal{D}'^*(\theta) \in \arg\max_{\mathcal{D}' \in \mathcal{U}(\mathcal{D}_n)} \mathbb{E}_{\mathcal{D}'}[L(\theta^*, z)]$ for any $\theta$, for simplicity we can assume $\mathcal{D}'^*(\theta)$ is unique since the supremum is unique. We consider the minimization problem: $\inf_{\theta \in \Theta} \mathbb{E}_{\mathcal{D}'^*(\theta)}[L(\theta, z)]$. Furthermore, the expected loss objective $\mathbb{E}_{Z \sim \mathcal{D}'}[L(\theta, z)]$ inherits lower-semicontinuity in $\theta$ from the loss function $L(\theta, z)$. Specifically, lower semi-continuity holds because

$$\liminf_{\theta_n \to \theta} \mathbb{E}_{Z \sim \mathcal{D}'}[L(\theta_n, Z)] \geq \mathbb{E}_{Z \sim \mathcal{D}'} \liminf_{\theta_n \to \theta}[L(\theta, z)] \geq \mathbb{E}_{Z \sim \mathcal{D}'}[L(\theta, z)],$$

then we have

$$\liminf_{\theta_n \to \theta} \mathbb{E}_{\mathcal{D}'^*(\theta)}[L(\theta_n, Z)] \geq \mathbb{E}_{\mathcal{D}'^*(\theta)}[L(\theta, z)]$$

and by definition of best response:

$$\liminf_{\theta_n \to \theta} \mathbb{E}_{\mathcal{D}'^*(\theta_n)}[L(\theta_n, Z)] \geq \liminf_{\theta_n \to \theta} \mathbb{E}_{\mathcal{D}'^*(\theta)}[L(\theta_n, Z)],$$

thus we have

$$\liminf_{\theta_n \to \theta} \mathbb{E}_{\mathcal{D}'^*(\theta_n)}[L(\theta_n, Z)] \geq \mathbb{E}_{\mathcal{D}'^*(\theta)}[L(\theta, z)],$$

which means the function $\inf_{\theta \in \Theta} \mathbb{E}_{\mathcal{D}'^*(\theta)}[L(\theta, z)]$ is lower-semi continuous on $\theta$. Finally, since $\Theta$ is compact, the minimization problem $\inf_{\theta \in \Theta} \mathbb{E}_{\mathcal{D}'^*(\theta)}[L(\theta, z)]$ has a solution $\theta^*$. $\qquad\square$

**Proof of the Minimax Theorem 11**  (Restate) If the loss function satisfies Assumption 8 holds, $\Theta$ is convex, and $L(\theta, z)$ is convex in $\theta$ for any $z \in \mathcal{Z}$, then we have

$$\min_{\theta \in \Theta} \max_{\mathcal{D}' \in \mathcal{U}(\mathcal{D}_n)} \mathbb{E}_{\mathcal{D}'}[L(\theta, z)] = \max_{\mathcal{D}' \in \mathcal{U}(\mathcal{D}_n)} \min_{\theta \in \Theta} \mathbb{E}_{\mathcal{D}'}[L(\theta, z)]. \tag{14}$$

*Proof.* We will verify the conditions of Sion's minimax theorem (Sion, 1958). First, we show that the ambiguity set $\mathcal{U}(\mathcal{D}_n)$ is convex. Assume that $\mathcal{D}^1, \mathcal{D}^2 \in \mathcal{U}(\mathcal{D}_n)$. Then there exist $\widehat{\mathcal{D}^i}(i = 1, 2) \in \mathcal{P}(\mathcal{Z})$ such that $W_p(\mathcal{D}_n, \widehat{\mathcal{D}^i}) \leq \varepsilon$, $KL(\widehat{\mathcal{D}^i}, \mathcal{D}^i) \leq \gamma$. As KL-divergence is convex, for any $\lambda \in [0, 1]$, we have

$$KL(\lambda\widehat{\mathcal{D}^1} + (1 - \lambda)\widehat{\mathcal{D}^2}, \lambda\mathcal{D}^1 + (1 - \lambda)\mathcal{D}^2) \leq \lambda KL(\widehat{\mathcal{D}^1}, \mathcal{D}^1) + (1 - \lambda)KL(\widehat{\mathcal{D}^2}, \mathcal{D}^2) \leq \gamma.$$

And for Wasserstein we have

$$W_p(\mathcal{D}_n, \lambda\widehat{\mathcal{D}^1} + (1 - \lambda)\widehat{\mathcal{D}^2}) \leq \lambda W_p(\mathcal{D}_n, \widehat{\mathcal{D}^1}) + (1 - \lambda)W_p(\mathcal{D}_n, \widehat{\mathcal{D}^2}) \leq \varepsilon.$$

Let $\mathcal{D}'' = \lambda\widehat{\mathcal{D}^1} + (1 - \lambda)\widehat{\mathcal{D}^2}$, which means $\lambda\mathcal{D}^1 + (1 - \lambda)\mathcal{D}^2 \in \mathcal{U}(\mathcal{D}_n)$. Now we have $\Theta$ is convex and $\mathcal{U}(\mathcal{D}_n)$ is convex and weak compact by proposition 9.

Furthermore, the expected loss objective $\mathbb{E}_{Z \sim \mathcal{D}'}[L(\theta, z)]$ inherits convexity and lower-semicontinuity in $\theta$ from the loss function $L(\theta, z)$. Specifically, lower semi-continuity holds because

$$\liminf_{\theta_n \to \theta} \mathbb{E}_{Z \sim \mathcal{D}'}[L(\theta_n, Z)] \geq \mathbb{E}_{Z \sim \mathcal{D}'} \liminf_{\theta_n \to \theta}[L(\theta, z)] \geq \mathbb{E}_{Z \sim \mathcal{D}'}[L(\theta, z)].$$

These inequalities leverage two key properties. The first inequality follows from Fatou's lemma for random variables with uniformly integrable negative parts, a property guaranteed by Assumption 8 (i) that loss function is non-negative. The second inequality exploits the lower semi-continuity of the loss function $L(\theta, \cdot)$ in $\theta$, as established in Assumption 8 (ii).

Finally, it is readily verified that the objective function $\mathbb{E}_{Z \sim \mathcal{D}'}[L(\theta, z)]$ is concave (in fact, linear) and weakly upper semi-continuous in $\mathcal{D}'$. This follows directly from the proof of Proposition 9. This analysis establishes that all the conditions of Sion's minimax theorem are satisfied. Consequently, the infimum and supremum in Equation (7) can be interchanged.

It remains to show that both maxima in (7) are reached. However, this follows immediately from the weak compactness of $\mathcal{U}(\mathcal{D}_n)$ and the weak upper semi-continuity in $\mathcal{D}'$ of both the expected loss $\mathbb{E}_{Z \sim \mathcal{D}'}[L(\theta, z)]$ and the optimal expected loss $\inf_{\theta \in \Theta} \mathbb{E}_{Z \sim \mathcal{D}'}[L(\theta, z)]$. $\qquad\square$

## A.5 PROOF OF PROPOSITION 12

In this subsection, we prove the dual formulation of the robust maximization problem.

**Proof of Strong Duality (Proposition 12)** (Restate) $\mathcal{L}_{\varepsilon,r}(\theta, \mathcal{D})$ admits the dual formulation for all $\gamma > 0$:

$$\mathcal{L}_{\varepsilon,\gamma}(\theta, \mathcal{D}) = \inf_{\substack{\lambda,\beta \geq 0 \\ \eta \geq \max_{z \in \mathcal{Z}} L(\theta,z)}} \left\{ \lambda \varepsilon^p + \mathbb{E}_{\mathcal{D}}[\varphi(\lambda, \beta, \eta, z)] \right\}$$

where $\varphi(\lambda, \beta, \eta, z) = \sup_{\xi \in \mathcal{Z}} \{\beta \log \frac{\beta}{\eta - L(\theta,\xi)} + (\gamma - 1)\beta + \eta - \lambda d^p(z,\xi)\}$.

*Proof.*

$$L_{\varepsilon,\gamma}(\theta, \mathcal{D})$$
$$=: \sup \left\{ \mathbb{E}_{\mathcal{D}'}[L(\theta, z)] : D', \mathcal{Q} \in \mathcal{P}(\mathcal{Z}), W_p(\mathcal{D}, \mathcal{Q}) \leq \varepsilon, \mathrm{KL}(\mathcal{Q}, \mathcal{D}') \leq \gamma \right\}$$
$$= \sup \left\{ \sup \left\{ \mathbb{E}_{\mathcal{D}'}[L(\theta, z)] : \mathcal{D}' \in \mathcal{P}(\mathcal{Z}), \mathrm{KL}(\mathcal{Q}, \mathcal{D}') \leq \gamma \right\} : \mathcal{Q} \in \mathcal{P}(\mathcal{Z}), W_p(\mathcal{D}, \mathcal{Q}) \leq \varepsilon \right\}$$

$$\overset{(1)}{=} \sup_{\substack{\mathcal{Q} \in \mathcal{P}(\mathcal{Z}), \\ W_p(\mathcal{D},\mathcal{Q}) \leq \varepsilon}} \left\{ \inf_{\substack{\beta \geq 0 \\ \eta \geq \max_{z \in \mathcal{Z}} L(\theta,z)}} \left\{ \int \beta \cdot \log \frac{\beta}{\eta - L(\theta, z)} d\mathcal{Q}(z) + (\gamma - 1)\beta + \eta \right\} \right\}$$

$$\overset{(2)}{=} \lim_{\tau \to 0_+} \sup_{\substack{\mathcal{Q} \in \mathcal{P}(\mathcal{Z}), \\ W_p(\mathcal{D},\mathcal{Q}) \leq \varepsilon}} \left\{ \inf_{\substack{\beta \geq 0 \\ \eta \geq \max_{z \in \mathcal{Z}} L(\theta,z)+\tau}} \left\{ \int \beta \cdot \log \frac{\beta}{\eta - L(\theta, z)} d\mathcal{Q}(z) + (\gamma - 1)\beta + \eta \right\} \right\}$$

$$\overset{(3)}{=} \lim_{\tau \to 0_+} \left\{ \inf_{\substack{\beta \geq 0 \\ \eta \geq \max_{z \in \mathcal{Z}} L(\theta,z)+\tau}} \left\{ \sup_{\substack{\mathcal{Q} \in \mathcal{P}(\mathcal{Z}), \\ W_p(\mathcal{D},\mathcal{Q}) \leq \varepsilon}} \int \beta \cdot \log \frac{\beta}{\eta - L(\theta, z)} d\mathcal{Q}(z) + (\gamma - 1)\beta + \eta \right\} \right\}$$

$$= \lim_{\tau \to 0_+} \left\{ \inf_{\substack{\beta \geq 0 \\ \eta \geq \max_{z \in \mathcal{Z}} L(\theta,z)+\tau}} \left\{ \sup_{\substack{\mathcal{Q} \in \mathcal{P}(\mathcal{Z}), \\ W_p(\mathcal{D},\mathcal{Q}) \leq \varepsilon}} \mathbb{E}_{\mathcal{Q}} \left[ \beta \cdot \log \frac{\beta}{\eta - L(\theta, z)} + (\gamma - 1)\beta + \eta \right] \right\} \right\}$$

$$\overset{(4)}{=} \lim_{\tau \to 0_+} \left\{ \inf_{\substack{\beta \geq 0 \\ \eta \geq \max_{z \in \mathcal{Z}} L(\theta,z)+\tau}} \left\{ \inf_{\lambda \geq 0} \lambda \varepsilon^p + \mathbb{E}_{\mathcal{D}} \left[ \varphi(\lambda, \beta, \eta, z) \right] \right\} \right\}$$

$$= \inf_{\substack{\beta \geq 0, \lambda \geq 0 \\ \eta \geq \max_{z \in \mathcal{Z}} L(\theta,z)}} \lambda \varepsilon^p + \mathbb{E}_{\mathcal{D}} \left[ \varphi(\lambda, \beta, \eta, z) \right].$$

Here, the equality (1) follows from Lemma 21. The equality (2) follows from the fact that

$$\inf \left\{ \int \beta \log \left( \frac{\beta}{\eta - L(\theta, z)} \right) d\mathcal{Q}(z) + (\gamma - 1)\beta + \eta : \beta \geq 0, \eta \geq \max_{z \in \mathcal{Z}} L(\theta, z) \right\}$$
$$\leq \inf \left\{ \int \beta \log \left( \frac{\beta}{\eta - L(\theta, z)} \right) d\mathcal{Q}(z) + (\gamma - 1)\beta + \eta : \beta \geq 0, \eta \geq \max_{z \in \mathcal{Z}} L(\theta, z) + \tau \right\}$$
$$\leq \inf \left\{ \int \beta \log \left( \frac{\beta}{\eta - L(\theta, z)} \right) d\mathcal{Q}(z) + (\gamma - 1)\beta + \eta : \beta \geq 0, \eta \geq \max_{z \in \mathcal{Z}} L(\theta, z) \right\} + \tau.$$

for any $\tau > 0$. The equality (3) follows from the minimax theorem of (Sion, 1958), and we refer to (Theorem 3.4's proof (Bennouna & Van Parys, 2022)) for the proof of this part. Finally, equality (4) follows from the strong duality of Wasserstein strong duality. $\qquad \square$

**Lemma 21.** *(Van Parys et al., 2021) We have*

$$\sup_{\substack{\mathcal{Q} \in \mathcal{P}(\mathcal{Z}), \\ \mathrm{KL}(\mathcal{D}', \mathcal{Q}) \leq \gamma}} \mathbb{E}_{\mathcal{D}'}\left[L(\theta, z)\right] = \inf_{\substack{\beta \geq 0, \\ \eta \geq \max_{\xi \in \mathcal{Z}} L(\theta, \xi)}} \left\{ \int \beta \log\left(\frac{\beta}{\eta - L(\theta, \xi)}\right) \mathrm{d}\mathcal{D}'(\xi) + (\gamma - 1)\beta + \eta \right\}$$

*for all $\mathcal{D}' \in \mathcal{P}(\mathcal{Z})$ and $\gamma > 0$.*

**Lemma 22.** *(Blanchet & Murthy, 2019) For all $\mathcal{D}'$ we have:*

$$\sup_{\substack{\mathcal{Q} \in \mathcal{P}(\mathcal{Z}), \\ \mathrm{W}_p(\mathcal{Q}, \mathcal{D}') \leq \varepsilon}} \mathbb{E}_{\mathcal{Q}}\left[L(\theta, z)\right] = \inf_{\lambda \geq 0}\left\{\lambda \varepsilon^p + \mathbb{E}_{\mathcal{D}'}[\varphi(\lambda, z)]\right\}$$

*where $\varphi(\lambda, z) := \sup_{\xi \in \mathcal{Z}}\{L(\theta, \xi) - \lambda \mathrm{d}^p(z, \xi)\}$.*

## A.6 FURTHER DETAILS ON EXPERIMENTS

We use the ResNet-18 for the CIFAR-10 and CIFAR-100 dataset. We use the SGD optimizer with momentum 0.9, weight decay 5e-4. The starting learning rate is 0.1 and reduce the learning rate ($\times 0.1$) at epoch $\{100, 150\}$. We train with 200 epochs.

**Mitigating overfitting on more attack budgets.** We would like to provide further experimental results on the CIFAR-10 dataset on more attack budgets (training and test attack budget is same) as shown in Table 4. The gap between these final robust accuracy and best robust accuracy is the smallest in our method, indicating that our approach most effectively mitigates overfitting even for different attack budgets.

Table 4: Robust performance of different methods on CIFAR-10 using a ResNet-18 for $l_\infty$ with budget 6/255, 10/255 and 12/255. We use PGD-10 as the attack to evaluate robust performance. Nat is the natural test accuracy. The "Best" robust test acc is the highest robust test accuracy achieved during training whereas the "Final" robust test acc is last epoch's robust accuracy.

| Robust Methods | Nat | Robust Test Acc (%) | | |
|---|---|---|---|---|
| | | Final | Best | Diff |
| $\varepsilon = 6/255$ | | | | |
| PGD-AT | $\mathbf{87.46 \pm 0.17}$ | $54.33 \pm 0.12$ | $60.41 \pm 0.08$ | $6.08 \pm 0.18$ |
| UDR-AT | $86.59 \pm 0.04$ | $54.35 \pm 0.55$ | $\mathbf{60.85 \pm 0.16}$ | $6.51 \pm 0.41$ |
| HR | $87.02 \pm 0.12$ | $55.23 \pm 0.35$ | $58.67 \pm 0.12$ | $3.44 \pm 0.42$ |
| Ours | $86.36 \pm 0.23$ | $\mathbf{56.52 \pm 0.54}$ | $59.57 \pm 0.13$ | $\mathbf{3.06 \pm 0.66}$ |
| $\varepsilon = 8/255$ | | | | |
| PGD-AT | $\mathbf{84.80 \pm 0.14}$ | $45.16 \pm 0.19$ | $52.91 \pm 0.11$ | $7.75 \pm 0.17$ |
| UDR-AT | $83.87 \pm 0.26$ | $46.60 \pm 0.27$ | $\mathbf{53.23 \pm 0.30}$ | $6.63 \pm 0.57$ |
| HR | $83.95 \pm 0.32$ | $47.32 \pm 0.59$ | $51.23 \pm 0.25$ | $3.90 \pm 0.47$ |
| Ours | $83.34 \pm 0.16$ | $\mathbf{48.58 \pm 0.21}$ | $51.95 \pm 0.19$ | $\mathbf{3.36 \pm 0.06}$ |
| $\varepsilon = 10/255$ | | | | |
| PGD-AT | $\mathbf{82.32 \pm 0.33}$ | $38.51 \pm 0.06$ | $46.45 \pm 0.22$ | $7.94 \pm 0.26$ |
| UDR-AT | $81.49 \pm 0.54$ | $39.54 \pm 0.64$ | $\mathbf{47.33 \pm 0.55}$ | $7.79 \pm 0.27$ |
| HR | $80.90 \pm 0.40$ | $41.78 \pm 0.43$ | $45.70 \pm 0.02$ | $3.93 \pm 0.41$ |
| Ours | $80.34 \pm 0.97$ | $\mathbf{43.52 \pm 0.59}$ | $46.47 \pm 0.13$ | $\mathbf{3.11 \pm 0.64}$ |
| $\varepsilon = 12/255$ | | | | |
| PGD-AT | $\mathbf{79.55 \pm 0.06}$ | $34.16 \pm 0.30$ | $41.79 \pm 0.12$ | $7.63 \pm 0.41$ |
| UDR-AT | $78.67 \pm 0.61$ | $35.93 \pm 1.06$ | $\mathbf{42.69 \pm 0.10}$ | $6.76 \pm 1.01$ |
| HR | $77.86 \pm 0.98$ | $37.69 \pm 0.61$ | $41.50 \pm 0.26$ | $3.81 \pm 0.76$ |
| Ours | $76.33 \pm 1.18$ | $\mathbf{39.45 \pm 0.58}$ | $42.24 \pm 0.31$ | $\mathbf{2.79 \pm 0.53}$ |

Table 5: Robust performance of different robust methods based on TRADES on CIFAR-10 using a ResNet-18 for $l_\infty$ with budget 8/255. We use PGD-10 as the attack to evaluate robust performance. The "Best" robust test acc is the highest robust test accuracy achieved during training whereas the "Final" robust test acc is last epochs's robust accuracy.

| Robust Methods | Nat | Robust Test Acc (%) | | |
| --- | --- | --- | --- | --- |
| | | Final | Best | Diff |
| TRADES | $82.69 \pm 0.32$ | $51.07 \pm 0.18$ | $53.54 \pm 0.17$ | $2.47 \pm 0.21$ |
| UDR-TRADES | $\mathbf{82.74 \pm 0.59}$ | $51.16 \pm 0.53$ | $\mathbf{53.64 \pm 0.15}$ | $2.48 \pm 0.69$ |
| Ours-TRADES | $81.79 \pm 0.55$ | $\mathbf{52.22 \pm 0.42}$ | $53.58 \pm 0.19$ | $\mathbf{1.36 \pm 0.62}$ |

Table 6: Robustness evaluation on CIFAR-10 using ResNet-18 for $l_\infty$ under attack budget 8/255, where models are trained with a larger attack budget 10/255.

| Methods(10/255) | Nat | PGD-10 | PGD-200 | AA |
| --- | --- | --- | --- | --- |
| PGD-AT | $\mathbf{82.32 \pm 0.33}$ | $46.98 \pm 0.06$ | $45.03 \pm 0.12$ | $43.17 \pm 0.08$ |
| UDR-AT | $81.49 \pm 0.54$ | $48.12 \pm 0.41$ | $46.45 \pm 0.63$ | $43.85 \pm 0.57$ |
| HR | $80.90 \pm 0.40$ | $50.04 \pm 0.45$ | $48.69 \pm 0.41$ | $44.05 \pm 0.42$ |
| Ours | $80.58 \pm 0.98$ | $\mathbf{50.86 \pm 0.72}$ | $\mathbf{49.71 \pm 0.88}$ | $\mathbf{45.86 \pm 0.88}$ |

Table 7: Robustness evaluation on CIFAR-10 using ResNet-18 for $l_\infty$ under attack budget 8/255, where models are trained with attack budget 12/255.

| Methods(12/255) | Nat | PGD-10 | PGD-200 | AA |
| --- | --- | --- | --- | --- |
| PGD-AT | $\mathbf{79.55 \pm 0.06}$ | $48.99 \pm 0.38$ | $47.20 \pm 0.26$ | $44.48 \pm 0.07$ |
| UDR-AT | $78.67 \pm 0.61$ | $50.38 \pm 0.63$ | $49.10 \pm 0.75$ | $45.47 \pm 0.72$ |
| HR | $76.65 \pm 0.65$ | $51.96 \pm 0.05$ | $51.03 \pm 0.12$ | $45.45 \pm 0.44$ |
| Ours | $76.33 \pm 1.17$ | $\mathbf{52.32 \pm 0.23}$ | $\mathbf{51.52 \pm 0.26}$ | $\mathbf{46.87 \pm 0.17}$ |

**Mitigating overfitting on TRADES.** We would like to provide further experimental results on the CIFAR-10 dataset on TRADES and its counterpart as shown in Table 5. It can be observed that our method also can mitigates the robust overfitting, and enhances robustness while maintaining comparable accuracy.

**Training with larger budgets.** Theorem 5 shows that to ensure robustness on the test set, it is generally necessary to employ a larger attack budget during training compared to testing. In this experiment, we examine the robustness generalization by attacking different robust methods with fixed attack budget 8/255 with respect to larger training budgets 10/255, 12/255 while keeping other parameters of PGD attack the same. The results shown in Table 6 and Table 7 demonstrate that our proposed SR-WDRO achieves the best robustness under various adversarial attacks.

**Computation cost** We evaluate the computational efficiency of our proposed SR-WDRO method against baseline approaches. Table presents the average training time per epoch and total training time for CIFAR-10 on a single NVIDIA A800 GPU. Our SR-WDRO incurs a modest increase in per-epoch time (approximately 12.7% longer than standard adversarial training).

Table 8: Training time per epoch and total training time for CIFAR-10 on a single NVIDIA A800 GPU.

| Time Cost (s) | PGD-AT | UDR-AT | HR | Ours |
|---|---|---|---|---|
| 200 epoch | 12614.59 | 14104.05 | 13705.56 | 14216.43 |
| per epoch | 63.07 | 70.52 | 68.53 | 71.08 |

