# OpenReview forum: "Provable Robust Overfitting Mitigation in Wasserstein Distributionally Robust Optimization"
_ICLR.cc/2025/Conference — ICLR 2025 Poster_

### Official Review · Reviewer_DJPB · 2024-10-21

**Soundness:** 3
**Presentation:** 3
**Contribution:** 3
**Rating:** 6
**Confidence:** 4

**Summary:**

This paper studies the Wasserstein distributionally robust optimization (WDRO) and considers both the adversarial attack and statistical error in the framework. Additional discussions are also provided in terms of the Stackelberg and Nash equilibria.

**Strengths:**

(1) The paper is clear and easy to understand.

(2) The theoretical analysis is also sound.

(3) The Nash equilibria perspective is interesting.

**Weaknesses:**

(1) The numerical experiments only demonstrate limited improvements.

(2) There is no enough highlight on the technical challenges.

(3) Based on existing litertaure,

Li, Binghui, and Yuanzhi Li. "Why clean generalization and robust overfitting both happen in adversarial training." (2023).

the robust overfitting phenomenon is more severe in the scenario of neural networks. Could the authors point out any possible way of analyzing WDRO in neural networks?

**Questions:**

Please address my comments in the weakness section.

---

> ### Author Response · Authors · 2024-11-22
> **# Response to Reviewer DJPB**
>
> We appreciate the thoughtful questions and comments of the reviewer. Below are our responses to your queries:
>
> ---
>
> ### 1. Limited Improvements in Numerical Experiments
>
> We would like to emphasize that our method is currently one of the most effective in mitigating robust overfitting. Furthermore, while addressing robust overfitting, our method still achieves the highest robust accuracy, demonstrating its effectiveness in balancing both aspects. We believe that with further research, particularly in improving the solution methods for SR-WDRO, even greater advancements can be achieved. We are committed to refining our approach and further enhancing its performance in future work.
>
> ---
>
> ### 2. Highlighting Technical Challenges
>
> We appreciate feedback on the need to better emphasize the technical challenges addressed in our work. We would like to highlight several key technical challenges of our work:
>
> (1). **Generalization Theorems:**
>    Firstly, in proving our generalization theorems in Section 4.2, we faced the challenge of not being able to rely on the dual form of WDRO as in previous works [1,2]. Instead, we employed the Large Deviation Principle to demonstrate that the test distribution falls within our defined ambiguity set with high probability. This was a significant shift from conventional methodologies and is fundamentally different from the standard WDRO.
>
>    More specifically, the conditions required for the generalization bound in [2] differ significantly from ours. In particular, the results in [2] rely on several assumptions about the loss function, including its Lipschitz continuity and boundedness, which are not required in our framework. Furthermore, the form of the generalization bounds in [2] is entirely different from ours.  Our bound provides a generalization guarantee where the probability of failure (i.e., the complement of the confidence level) decays exponentially with the sample size $n$. Moreover, the rate of decay is directly influenced by $\gamma$.
>
> - **Our Bound (Theorem 3):**
>
> $$P(E_{D}[L(\theta, z)] \leq \mathcal{L}_{\varepsilon, \gamma}(\theta, D_n) )\geq 1 - e^{-\gamma n} \left(\frac{4}{\delta}\right)^{m(\mathcal{Z}, \delta)}, \quad  \text{where} \quad \delta = (\frac{\varepsilon}{diam(Z)+1})^p.$$
>
> - **WDRO’s Bound (Theorem 3.1 in [2]):**
>
> $$
> P(E_{D}[L(\theta, z)] \leq L_{\varepsilon, 0}(\theta, D_n) ) \geq 1 - \delta,
> $$
>
> where $\varepsilon$ must satisfy:
>
> $$
> \mathcal{O}\left(\sqrt{\frac{1 + \log(1/\delta)}{n}}\right) \leq \varepsilon \leq \frac{\varepsilon_c}{2} - \mathcal{O}\left(\sqrt{\frac{1 + \log(1/\delta)}{n}}\right),
> $$
>
> and $\varepsilon_c$ is a constant that depends only on the loss function and the data distribution $D$. The computation of this bound requires the dual form of WDRO.
>
>
> (2). **Game Equilibrium:**
>    Secondly, establishing the existence of the game equilibrium in Section 4.3 was another technical hurdle. We had to prove the compactness of our newly defined ambiguity set and verify certain continuity conditions of the loss function. These proofs were nontrivial. Furthermore, the Stackelberg equilibrium is first established by us in the WDRO regime, which requires essentially a weaker condition than that of Nash.
>
> ---
> ### 3. Analyzing WDRO in Neural Networks
>
> Our framework is designed to be versatile, with $\theta$ capable of representing any model type, including neural networks. In our exploration of adversarial distribution shifts, $\theta$ specifically denotes neural networks, such as ResNet. This enables us to directly apply and analyze our SR-WDRO approach within the context of neural networks, effectively addressing the phenomenon of robust overfitting.
>
> ---
>
> #### References
>
> [1] An, Y., Gao, R. (2021). Generalization bounds for (Wasserstein) robust optimization. Advances in Neural Information Processing Systems, 34, 10382-10392.
>
> [2] Azizian, W., Iutzeler, F., Malick, J. (2023). Exact generalization guarantees for (regularized) wasserstein distributionally robust models. Advances in Neural Information Processing Systems, 36, 14584-14596.

---

> > ### Comment · Reviewer_DJPB · 2024-11-22
> >
> > Thank you for your response. I'm wondering if the authors have time to try using generated data in https://arxiv.org/abs/2110.09468 ? In this paper, they use WideResNet28-10 and generated data and obtain 60% robust accuracy. Their data are available in github as well. Since in https://robustbench.github.io/ , most methods are using generated data, and the authors want to sell the effectiveness of their method as a notable contribution, comparing with sota methods would be beneficial.
> >
> > Otherwise I will keep my score. I recognize the theoretical contribution of this paper.

---

> > > ### Author Response · Authors · 2024-11-27
> > >
> > > We sincerely thank the reviewer for their thoughtful feedback and for acknowledging the theoretical contributions of our work. We deeply appreciate the time and effort you have dedicated to providing such constructive suggestions.
> > >
> > > We also greatly value your insightful comments regarding comparisons with newer approaches, including those using generated data. We will try our best to incorporate such comparisons to further validate and strengthen the contributions of our work.
> > >
> > > Thank you once again for your valuable feedback and support!

---

### Official Review · Reviewer_cser · 2024-11-03

**Soundness:** 3
**Presentation:** 3
**Contribution:** 3
**Rating:** 8
**Confidence:** 3

**Summary:**

The paper proposes a novel framework for the problem of Wasserstein distributionally robust optimization (WDRO) by introducing a novel ambiguity set. The effectiveness of the proposed approach is backed by the generalization certificate bound and the authors also establish the existence for the Stackelberg and Nash equilibria of the statistically robust WDRO problem. Empirically, the proposed practical training algorithm demonstrates its advantages on different adversarial robustness benchmarks.

**Strengths:**

The proposed method seems theoretically sound,  and its effectiveness is supported by different empirical experiments. The authors offer a comprehensive analysis that encompasses both theoretical insights and empirical validations. I commend the authors for their commitment to reproducibility by making their code publicly available. Additionally, the detailed explanations of the experimental setup and the thoughtful interpretation of the results are particularly noteworthy.

**Weaknesses:**

- The experiments appear to be limited in scope: (i) they only compare against older baselines, and (ii) the model architecture and datasets used are relatively small-scale.
- Empirically, the gains in adversarial robustness seem to come at the expense of natural accuracy, as observed in Table 1.

**Questions:**

- Step 10 in Algorithm 1 is not clear to me. How can we compute the optimal weights {pi}?
- Why use $\operatorname{sign}\left(\nabla_x L\left(\theta,\left(x_i^{k-1}, y_i\right)\right)\right)$ to update the adversarial examples instead of $\left(\nabla_x L\left(\theta,\left(x_i^{k-1}, y_i\right)\right)\right)$?
- A related work [1] that also incorporates both local and global information to optimize distributional robustness is worth discussing.
- As distributional robustness is known to address natural distributional shifts, how well do you expect the method to perform under such circumstances (e.g., domain adaptation/generalization)?

[1] Phan, Hoang, et al. "Global-local regularization via distributional robustness." International Conference on Artificial Intelligence and Statistics. PMLR, 2023.

Minor: Typo in line 104

---

> ### Author Response · Authors · 2024-11-22
> **# Response to Reviewer cser**
>
> We sincerely appreciate the reviewer's thoughtful questions and constructive feedback. Below, we provide our responses, carefully addressing both weaknesses and specific questions.
>
> ---
>
> ### **Addressing Weaknesses**
>
> 1. **Scope of Experiments**
>    We appreciate the reviewer's observations regarding the scope of our experiments. In response, we have expanded our experimental section to include comparisons with the work GLOT by Phan et al. [1], as suggested. As shown in the following table, our experiments on CIFAR10 with ResNet18 demonstrate that while GLOT reduces adversarial overfitting through global and local regularization, our method, SR-WDRO, achieves substantially better robustness on the test set, with an improvement of about 3% on CIFAR-10.
>
> | Robust Methods   | Nat Accuracy (%)        | Final Robust Test Acc (%) | Best Robust Test Acc (%) | Diff (%)            |
> |------------------|-------------------------|---------------------------|---------------------------|---------------------|
> | PGD-AT           | 84.80 ± 0.14           | 45.16 ± 0.19              | 52.91 ± 0.11              | 7.75 ± 0.17         |
> | UDR-AT           | 83.87 ± 0.26           | 46.60 ± 0.27              | **53.23 ± 0.30**          | 6.63 ± 0.57         |
> | HR               | 83.95 ± 0.32           | 47.32 ± 0.59              | 51.23 ± 0.25              | 3.90 ± 0.47         |
> | **GLOT**         | **86.08 ± 0.19**       | 45.23 ± 0.60              | 48.67 ± 0.32              | 3.44 ± 0.56         |
> | **Ours**         | 83.34 ± 0.16           | **48.58 ± 0.21**          | 51.95 ± 0.19              | **3.36 ± 0.06**     |
>
>
>    Additionally, we have conducted experiments using larger network architectures WideResNet28-10 on CIFAR10 to further validate the scalability and robustness of our method. These results show that our approach consistently outperforms the baselines and maintains strong performance across various architectures, as shown in the following table.
> | Robust Methods   | Nat Accuracy (%)        | Final Robust Test Acc (%) | Best Robust Test Acc (%) | Diff (%)            |
> |------------------|-------------------------|---------------------------|---------------------------|---------------------|
> | PGD-AT           | 86.51 ± 0.16           | 48.81 ± 0.49              | 55.64 ± 0.07              | 6.83 ± 0.55         |
> | UDR-AT           | 85.99 ± 0.15           | 49.01 ± 0.13              | **55.82 ± 0.19**          | 6.81 ± 0.31         |
> | HR               | 84.89 ± 0.16           | 48.45 ± 0.31              | 52.45 ± 0.27              | 4.00 ± 0.34         |
> | GLOT             | **86.57 ± 0.22**       | 49.45 ± 0.32              | 52.26 ± 0.46              | 2.81 ± 0.15         |
> | Ours             | 84.52 ± 0.27           | **51.26 ± 0.42**          | 53.87 ± 0.06              | **2.61 ± 0.47**     |
>
>
> 2. **Trade-off Between Adversarial Robustness and Natural Accuracy**
>    Regarding the observed trade-off between adversarial robustness and natural accuracy, we acknowledge this issue, as highlighted in Table 1. While this trade-off exists, our approach is notably effective in mitigating robust overfitting during training, which we consider a critical achievement. Robust overfitting is a significant challenge in adversarial training, and our method explicitly addresses this challenge. We acknowledge the importance of balancing robustness and accuracy and are committed to exploring further methods to address this trade-off more effectively in future work.
>
> [1] Phan, H., et al. (2023). Global-local regularization via distributional robustness. *International Conference on Artificial Intelligence and Statistics (AISTATS)*.

---

> ### Author Response · Authors · 2024-11-22
>
> ### **For Questions**
>
> #### **1. Step 10 in Algorithm 1**
>
> According to Eq.(9), initially, we generate adversarial examples for a given batch. Once adversarial examples $\{(x_i', y_i)\} _ {i=1}^n$ are identified, the task of computing $\(p_i\)_{i=1}^n$ becomes a convex optimization problem. Specifically:
>
> - The objective is to maximize the weighted loss:
>
>   $$
>   \max \sum_{i=1}^n p_i L\left(\theta, {(x_i', y_i)}\right),
>   $$
>   subject to the following constraints:
>   - $(\sum_{i=1}^n p_i = 1\),$
>   - $(KL(q \| p) = \sum_{i=1}^n q_i \log \left(\frac{q_i}{p_i}\right) \leq \gamma\), $where $(q = \left(\frac{1}{n}, \cdots, \frac{1}{n}\right))$.
>
> Then, this optimization can be efficiently executed using standard solvers.
>
>
> #### **2. Updating Adversarial Examples**
> Updating Adversarial Examples: We use the sign function for updating adversarial examples due to its effectiveness in generating perturbations. The sign function, as employed in famous methods like FGSM[1] and PGD[2], ensures that perturbations are uniformly small, yet sufficient to cause misclassification, adhering to the constraint $||\eta||_\infty$.
>
> ---
>
> #### **3. Comparison with Phan et al. (GLOT)**
> We thank the reviewer for highlighting the related work by Phan et al. [3], which enhances distributional robustness through local and global regularization. This approach is different from ours: we use KL divergence to enlarge the ambiguity set, thereby explicitly addressing statistical error, while Phan et al. [3] use KL divergence as a regularization term, akin to the TRADES method [4]. We will discuss their method in our revised manuscript and include experimental comparisons (have mentioned before) to highlight the distinct strengths and contributions of each approach.
>
> ---
>
> #### **4. Performance Under Natural Distributional Shifts**
> Performance under Natural Distributional Shifts: Our framework is indeed designed to accommodate various types of distributional shifts, including but not limited to the adversarial examples that we primarily discuss. We can theoretically extend our robustness guarantees, similar to those in Theorem 4, to other distributional shifts, provided that they remain within a certain bounded distance.
>
> To address this point, we have included additional theoretical analysis and discussions in the appendix A.3 of our revised manuscript to clarify these robustness guarantees under broader conditions.
>
> ---
>
> ### **References**
>
> [1] Goodfellow, I. J., Shlens, J., & Szegedy, C. (2014). Explaining and harnessing adversarial examples. *arXiv preprint arXiv:1412.6572*.
>
> [2] Madry, A., Makelov, A., Schmidt, L., Tsipras, D., & Vladu, A. (2018). Towards deep learning models resistant to adversarial attacks. *ICLR*.
>
> [3] Phan, H., et al. (2023). Global-local regularization via distributional robustness. *International Conference on Artificial Intelligence and Statistics (AISTATS)*.
>
> [4] Zhang, H., Yu, Y., Jiao, J., Xing, E., El Ghaoui, L., & Jordan, M. (2019). Theoretically principled trade-off between robustness and accuracy. *International Conference on Machine Learning (ICML)*.

---

> > ### Comment · Reviewer_cser · 2024-11-26
> >
> > Thank you for the response, the authors have effectively resolved all major points I raised. After reading the rebuttal and other reviews, I still believe that authors should conduct experiment on more up-to-date benchmark (e.g. RobustBench) and compare with recently proposed methods. Given the paper contribution from theoretical aspect and the effort of the response, I will update my score and vote for acceptance.

---

> > > ### Author Response · Authors · 2024-11-27
> > >
> > > We are grateful that the score has been increased and sincerely appreciate the significant time and effort you dedicated to reviewing our paper. We also deeply value your thoughtful questions regarding the theoretical aspects of our work, recognizing their importance. We will certainly keep them in mind as we pursue future research directions.
> > >
> > > Thank you once again for your valuable feedback and support!

---

### Official Review · Reviewer_CBH1 · 2024-11-03

**Soundness:** 2
**Presentation:** 2
**Contribution:** 2
**Rating:** 6
**Confidence:** 4

**Summary:**

The paper proposes a novel approach called SR-WDRO to address robust overfitting in Wasserstein Distributionally Robust Optimization (WDRO) by incorporating Kullback-Leibler (KL) divergence. While the theoretical contributions are interesting, there are several concerns regarding clarity and experimental validation.

**Strengths:**

**Theory:**

The mathematical framework is well-developed with thorough theoretical analysis. This paper provides two sets of theoretical results.

The first one is the generalization/robustness bound. The main idea is to show a high probability bound on D ∈ U(D_n) when the uncertainty set is defined with both two divergences. The second one is to establish the Stackelberg and Nash equilibria.

**Experiments:**

The proposed method demonstrates some improvement in mitigating robust overfitting compared to UDR and HR.

**Weaknesses:**

**Clarity Issue:**

The main idea of the paper is summarized in Line 169: "To mitigate this issue, we incorporate the Kullback-Leibler (KL) divergence in WDRO, specifically aiming to reduce statistical error caused by training on finite samples." However, this sentence lacks clarity in three critical aspects:

1. What is the definition of statistical error?

The term "statistical error" appears multiple times starting from the abstract. However, while "statistical" and "error" are very general terms, it is hard to understand what this refers to exactly in the mathematical framework.

2. Why is statistical error caused by training on finite samples?

Without a clear definition of statistical error, it is difficult to understand or verify this claim.

3. Why does incorporating the Kullback-Leibler (KL) divergence in WDRO mitigate this issue?

In the rest of the paper, neither the theoretical results, including the bounds and Nash equilibrium, nor the experimental results provide a clear answer to this question.

**Major Theoretical Concern:**

Sections 3 and 4 demonstrate that SR-WDRO possesses good bounds and equilibria properties. However, the paper does not discuss whether WDRO has or lacks these properties. Without this comparison, it is difficult to verify the necessity of introducing the SR- prefix.

For example, in Theorem 3, one could perform a simple sanity check:

>By letting γ=0 (which reduces SR-WDRO to WDRO), the generalization bound reduces to $P()\geq0$, which is a meaningless trivial bound. This suggests SR-WDRO has a better generalization bound than WDRO.

However, the authors should provide a deeper analysis than my simple observation.

The same question applies to Section 4 - do WDRO admit Stackelberg and Nash equilibria under similar assumptions? Without addressing these comparative aspects, the theoretical advantages of SR-WDRO over WDRO remain unclear.

**Major Experiments Concern:**

1. Regarding robust overfitting, while SR-WDRO outperforms UDR and HR, Figure 2 still shows a decreasing phase. This raises concerns about whether WDRO-type methods are truly necessary in adversarial training settings, especially considering that simpler approaches like SWA or EMA could mitigate the decreasing phase with better performance.

2. The comparison of the WDRO-type approaches with other types of adversarial training methods is not provided. As far as I know, it is not competitive with other methods under similar setting, such as no additional data, no generative data, and on ResNet-18.

3. The exclusive use of ResNet-18 is limiting, as the adversarial training community typically requires evaluation on larger models.

To summarize, while SR-WDRO appears to achieve the ceiling performance among WDRO-type algorithms, this result actually raises pessimism about applying WDRO to adversarial training. Despite WDRO's importance in other operations research problems, its effectiveness in adversarial training remains questionable.

**Questions:**

**Minor:**

1.	Eq. (1) and Line 38: Eq.(1) is defined directly in Wasserstein distance rather than the uncertain set. So the description of U(D_n) in line 38 is not self-consistent.

2.	The range of gamma in Theorem 3 is not stated.

3.	Theorem 3, line 204: internal covering number of Z. Line 210: covering number of Z. Line 745: internal covering number of A and covering number of Z. Since the definition with and without internal is different, please clarify these statement precisely.

---

> ### Author Response · Authors · 2024-11-22
> **Response to Reviewer CBH1**
>
> We sincerely appreciate the reviewer's feedback. We will address your concerns in three sections: Clarity Issue, Theoretical Concern, and Experiments Concern. Additionally, we have corrected the minor issues you pointed out in the revised version of our paper. Thank you for your valuable insights.
>
> ---
>
> ### 1. Clarity Issues
> To address the reviewer's concerns regarding clarity and the definition of statistical error, we offer the following explanations.
>
> 1. **Definition of Statistical Error:**
>    In our paper, the statistical error refers to the discrepancy between the finite sample data used during training and the true underlying data distribution, which was also used in this way in [1, 2]. This error arises because the limited dataset can only approximate the full distribution.
>
> 2. **Statistical Error from Finite Samples:**
>    Training on finite samples inherently introduces statistical error due to the sampling randomness and limited size of the training dataset. As a result, the empirical loss only serves as an approximation of the true out-of-sample loss and is subject to fluctuations caused by the randomness of the finite sample. This discrepancy can lead to scenarios where solutions appear effective in-sample but fail to generalize well out-of-sample, resulting in the phenomenon of overfitting.
>
> 3. **Mitigating Statistical Error with KL Divergence in WDRO:**
>    Building on the exploration of KL divergence's advantages in addressing statistical error, as demonstrated in Van Parys et al. [3], the KL-DRO predictor efficiently guards against overfitting by considering the worst-case expectation across a set of distributions, we have incorporated KL-DRO into WDRO framework. This integration helps to guard against robust overfitting and improves generalization by considering a range of distributions that include the true out-of-sample distributions with high probability.
>
> **References**:
> [1] Bennouna, A., Van Parys, B. (2022). Holistic robust data-driven decisions.
>
> [2] Bennouna, A., Lucas, R., Van Parys, B. (2023, July). Certified robust neural networks: Generalization and corruption resistance. In *International Conference on Machine Learning* (pp. 2092-2112). PMLR.
>
> [3] Van Parys, B. P., Esfahani, P. M., Kuhn, D. (2021). From data to decisions: Distributionally robust optimization is optimal. *Management Science*, 67(6), 3387-3402.

---

> ### Author Response · Authors · 2024-11-22
>
> ### 2. Theoretical Concerns
> We appreciate the reviewer's insightful comments and the opportunity to clarify the theoretical distinctions between SR-WDRO and WDRO.
>
> 1. **Comparison to WDRO:**
>   Our framework fundamentally distinguishes itself from standard WDRO by requiring $\gamma > 0$ to explicitly account for statistical error. In contrast, standard WDRO corresponds to the case where $\gamma = 0$, as the reviewer mentioned, the bound becomes $P(\cdot)$ is greater than a large negative, which is meaningless. However, as indicated in our experimental results, standard WDRO often suffers from robust overfitting due to its inability to effectively address statistical error. By setting $\gamma > 0$, our method mitigates this issue and enhances robustness, representing a significant shift from conventional WDRO approaches.
>   From a theoretical perspective, our framework differs from standard WDRO in both the assumptions required and the form of the generalization bounds:
>
> - **Assumptions:**
>   The generalization bounds in [2] rely on additional assumptions about the loss function, such as Lipschitz continuity and boundedness, which are not required in our framework.
>
> - **Form of the Bounds:**
>   The bounds in our framework are fundamentally different from those of WDRO. Our bound provides a generalization guarantee where the probability of failure (i.e., the complement of the confidence level) decays exponentially with the sample size $n$. Moreover, the rate of decay is directly influenced by $\gamma$.
>
>   For clarity, we summarize the key differences below:
>
>    #### **Our Bound (Theorem 3):**
>
>   $$P(E_{D}[L(\theta, z)] \leq \mathcal{L}_{\varepsilon, \gamma}(\theta, D_n) )\geq 1 - e^{-\gamma n} \left(\frac{4}{\delta}\right)^{m(\mathcal{Z}, \delta)}, \quad  \text{where} \quad \delta = (\frac{\varepsilon}{diam(Z)+1})^p.$$
>
>   #### **WDRO’s Bound (Theorem 3.1 in [2]):**
>
>   $$
>   P(E_{D}[L(\theta, z)] \leq L_{\varepsilon, 0}(\theta, D_n) ) \geq 1 - \delta,
>   $$
>   where $\varepsilon$ must satisfy:
>
>   $$
>   \mathcal{O}\left(\sqrt{\frac{1 + \log(1/\delta)}{n}}\right) \leq \varepsilon \leq \frac{\varepsilon_c}{2} - \mathcal{O}\left(\sqrt{\frac{1 +
>   \log(1/\delta)}{n}}\right),
>   $$
>
>   and $\varepsilon_c$ is a constant that depends only on the loss function and the data distribution $D$. The computation of this bound requires the dual form of WDRO.
>
> 2. **Existence of Game Equilibria:**
>    In response to the reviewer's question regarding Section 4, we acknowledge that under similar assumptions, WDRO can indeed admit Nash equilibria when viewed as a game-theoretical problem [3]. However, proving the existence of equilibria in SR-WDRO is more complex due to the introduction of a new uncertainty set. Furthermore, the existence of Stackelberg equilibrium (needs a weaker condition than that of Nash equilibrium) is first considered in this paper.
>
> **References**:
> [1] An, Y., Gao, R. (2021). Generalization bounds for (Wasserstein) robust optimization. *Advances in Neural Information Processing Systems*, 34, 10382-10392.
>
> [2] Azizian, W., Iutzeler, F., Malick, J. (2023). Exact generalization guarantees for (regularized) wasserstein distributionally robust models. *Advances in Neural Information Processing Systems*, 36, 14584-14596.
>
> [3] Shafieezadeh-Abadeh, S., Aolaritei, L., Dörfler, F., Kuhn, D. (2023). New perspectives on regularization and computation in optimal transport-based distributionally robust optimization.

---

> ### Author Response · Authors · 2024-11-22
>
> ### 3. Experiments Concerns
> We appreciate the reviewer's detailed feedback and recognize the need to address these experimental concerns.
>
> 1. **Effectiveness of Our Method:**
>    It is worth emphasizing that our method is currently among the most effective in mitigating robust overfitting, as demonstrated by our experimental results. We would like to further highlight that WDRO-type methods are indeed necessary in adversarial training settings. Prior works [1, 2] have extensively validated the advantages of WDRO-type approaches, both empirically and theoretically, demonstrating their strong generalization guarantees. Similar to adversarial training, WDRO-type methods serve as a foundational framework for robust training. Additionally, as you have suggested, techniques such as SWA or EMA can be incorporated into WDRO-based approaches to further enhance performance. We appreciate this suggestion and believe it highlights the potential for complementary improvements in future work.
>
> 2. **SR-WDRO as a Fundamental Framework:**
>    We would like to emphasize that our method, similar to adversarial training, serves as a fundamental framework for robust learning. Just as other types of methods have built upon adversarial training to achieve further improvements, these approaches can also be applied on top of SR-WDRO to enhance its performance. We agree that incorporating these advancements is a valuable direction, and we plan to explore such extensions in future work to further improve the effectiveness of SR-WDRO.
>
> 3. **Additional Experiments:**
>    We have conducted additional experiments using WideResNet28-10, a larger model commonly used in the adversarial training community. As shown in the following table, it can be observed that our method achieves the best performance in mitigating adversarial overfitting, and it achieves the highest robust accuracy, demonstrating the effectiveness of our approach on large models.
> | Robust Methods   | Nat Accuracy (%)        | Final Robust Test Acc (%) | Best Robust Test Acc (%) | Diff (%)            |
> |------------------|-------------------------|---------------------------|---------------------------|---------------------|
> | PGD-AT           | 86.51 ± 0.16           | 48.81 ± 0.49              | 55.64 ± 0.07              | 6.83 ± 0.55         |
> | UDR-AT           | 85.99 ± 0.15           | 49.01 ± 0.13              | **55.82 ± 0.19**          | 6.81 ± 0.31         |
> | HR               | 84.89 ± 0.16           | 48.45 ± 0.31              | 52.45 ± 0.27              | 4.00 ± 0.34         |
> | GLOT             | **86.57 ± 0.22**       | 49.45 ± 0.32              | 52.26 ± 0.46              | 2.81 ± 0.15         |
> | Ours             | 84.52 ± 0.27           | **51.26 ± 0.42**          | 53.87 ± 0.06              | **2.61 ± 0.47**     |
>
> **References**:
> [1] Bui, T. A., Le, T., Tran, Q., Zhao, H., Phung, D. (2022). A unified wasserstein distributional robustness framework for adversarial training.
>
> [2] Phan, Hoang, et al. Global-local regularization via distributional robustness. International Conference on Artificial Intelligence and Statistics. PMLR, 2023.

---

> > ### Comment · Reviewer_CBH1 · 2024-11-24
> > **Thanks for the response**
> >
> > I thank the authors for their comprehensive response.
> >
> > >1. Remarks 5 and 6 have addressed my questions regarding the theoretical aspects.
> >
> > >2. The experiments with WideResNet28-10 partially address my experimental concerns. I suggest including these WideResNet experiments in the main paper. Also, there are two consecutive periods ".." in line 1193 of the WideResNet experiments table that should be corrected.
> >
> > Given these improvements, I am increasing my score to 6. However, after reviewing papers [1,2], which compare their methods with baselines like AT and TRADES, I remain unconvinced about the significance of WDRO-type approaches for adversarial robustness without comparison to and/or combination with newer approaches (like those listed in Robustbench). Therefore, I cannot justify a higher score at this time.

---

> > > ### Author Response · Authors · 2024-11-27
> > >
> > > We sincerely thank the reviewer for their constructive feedback, acknowledgment of our work. We will include the WideResNet experiments in the main paper to provide a more comprehensive evaluation.
> > >
> > > We truly appreciate the reviewer’s thoughtful suggestions regarding comparisons with newer approaches, such as those in RobustBench. We will try our best to incorporate such comparisons to further validate and strengthen the contributions of our work.
> > >
> > > Thank you once again for your valuable feedback and support!

---

### Official Review · Reviewer_TYjK · 2024-11-09

**Soundness:** 3
**Presentation:** 3
**Contribution:** 3
**Rating:** 6
**Confidence:** 3

**Summary:**

This paper proposes a new method for distributional robust optimization by considering a relaxation of statistical error in the distribution sets

**Strengths:**

This paper proposes a new method for distributional robust optimization

**Weaknesses:**

see the questions below

**Questions:**

The paper is overall well written. I just have a couple of technical questions:
1. The result in Theorem 3 implicitly assumes that $e^{-\gamma\cdot n} (4/\delta)^m<1$, which is equivalently requiring $\gamma\ge\frac{m\log(4/\delta)}{n}$. As $m$ is the covering number, typically in the order of exp(d) in a d-dimensional space, this implies that $\gamma\ge \exp(d)/n$. Is this a too strict assumption?
2. It would be helpful to show in theory that the standard method without considering the statistical error relaxation fails, while the proposed method succeeds.
3. Is there a formal theory for the output of Algorithm 1?

---

> ### Author Response · Authors · 2024-11-22
> **# Response to Reviewer TYjK**
>
> We appreciate the reviewer’s thoughtful questions and insightful comments. Below, we address your concerns point by point:
>
> ---
>
> ### 1. Regarding the condition $\gamma > m(\mathcal{Z}, \delta) \cdot \log(4/\delta)/n$
>
> We understand that this condition might appear strong due to the exponential dependency on the dimensionality of the sample space $\mathcal{Z}$. However, it is important to note that the cover number is inherently not excessively large in practice because the intrinsic dimensionality of the sample space is often much lower than its ambient dimension. For instance, the intrinsic dimensionality of MNIST is around 13, CIFAR-10 approximately 26, and ImageNet between 26 and 43 ([1], [2]). This intrinsic dimension can be directly used in our theory, instead of the ambient dimension, which significantly reduces the burden of this condition in practical scenarios. We will clarify this distinction in the revised version of the paper, as discussed in Remark 5 of Section 4.2.
>
> Additionally, as the number of samples $n$ becomes sufficiently large, we can consistently find appropriate values of $\gamma$ that satisfy the condition. Empirically, we observe that properly chosen $\gamma$ values enhance generalization performance on the test set, further validating this condition in practice.
>
> ---
>
> ### 2. Comparison with standard WDRO
>
> Our framework fundamentally distinguishes itself from standard WDRO by requiring $\gamma > 0$ to explicitly account for statistical error. In contrast, standard WDRO corresponds to the case where $\gamma = 0$. However, as indicated in our experimental results, standard WDRO often suffers from robust overfitting due to its inability to effectively address statistical error. By setting $\gamma > 0$, our method mitigates this issue and enhances robustness, representing a significant shift from conventional WDRO approaches.
>
> From a theoretical perspective, our framework differs from standard WDRO in both the assumptions required and the form of the generalization bounds:
>
> - **Assumptions:**
>   The generalization bounds in [2] rely on additional assumptions about the loss function, such as Lipschitz continuity and boundedness, which are not required in our framework.
>
> - **Form of the Bounds:**
>   The bounds in our framework are fundamentally different from those of WDRO. Our bound provides a generalization guarantee where the probability of failure (i.e., the complement of the confidence level) decays exponentially with the sample size $n$. Moreover, the rate of decay is directly influenced by $\gamma$.
>
> For clarity, we summarize the key differences below:
>
>  #### **Our Bound (Theorem 3):**
>
> $$P(E_{D}[L(\theta, z)] \leq \mathcal{L}_{\varepsilon, \gamma}(\theta, D_n) )\geq 1 - e^{-\gamma n} \left(\frac{4}{\delta}\right)^{m(\mathcal{Z}, \delta)}, \quad  \text{where} \quad \delta = (\frac{\varepsilon}{diam(Z)+1})^p.$$
>
> #### **WDRO’s Bound (Theorem 3.1 in [2]):**
>
> $$
> P(E_{D}[L(\theta, z)] \leq L_{\varepsilon, 0}(\theta, D_n) ) \geq 1 - \delta,
> $$
>
> where $\varepsilon$ must satisfy:
>
> $$
> \mathcal{O}\left(\sqrt{\frac{1 + \log(1/\delta)}{n}}\right) \leq \varepsilon \leq \frac{\varepsilon_c}{2} - \mathcal{O}\left(\sqrt{\frac{1 + \log(1/\delta)}{n}}\right),
> $$
>
> and $\varepsilon_c$ is a constant that depends only on the loss function and the data distribution $D$. The computation of this bound requires the dual form of WDRO.
>
> ---
>
> ### 3. Formal theory for Algorithm 1
>
> Our method is grounded in the robust optimization framework, which seeks the most adversarial distribution within a specified ambiguity set defined by Wasserstein distance and KL divergence constraints. While we acknowledge the absence of a formal derivation using a strict dual formulation, we believe this represents an exciting direction for future work. Formalizing the duality theory for our approach would further solidify its theoretical foundation, and we plan to explore this in subsequent research.
>
> ---
>
> ### References
>
> [1] Facco, E., d’Errico, M., Rodriguez, A., & Laio, A. (2017). Estimating the intrinsic dimension of datasets by a minimal neighborhood information. *Scientific Reports*, 7(1), 12140.
>
> [2] Pope, P., Zhu, C., Abdelkader, A., Goldblum, M., & Goldstein, T. (2021). The Intrinsic Dimension of Images and Its Impact on Learning. *International Conference on Learning Representations (ICLR)*.

---

### Public Comment · ~Shuang_Liu5 · 2025-02-13

We thank the reviewers and the Area Chair for their valuable feedback during the discussion phase, which has significantly improved our paper. In the final camera-ready version, to better highlight our theoretical contributions, we have revised the title from "Mitigating Robust Overfitting in Wasserstein Distributionally Robust Optimization" to "Provable Robust Overfitting Mitigation in Wasserstein Distributionally Robust Optimization." We have also made further modifications throughout the paper to enhance its overall quality.

---

### Meta-Review · Area_Chair_YoJe · 2024-12-19

**Metareview:**

Wasserstein distributionally robust optimization (WDRO) improves generalization against adversarial examples by optimizing against worst-case distributional shifts, but it still faces robust overfitting because it doesn't account for statistical error. To address this, this paper proposes a new robust optimization framework that combines Wasserstein distance for adversarial noise and Kullback-Leibler divergence for statistical error. Theoretical analysis shows that the new framework improves out-of-distribution adversarial performance and ensures the existence of equilibrium between the learner and adversary, with experimental results demonstrating significant reductions in robust overfitting and enhanced robustness.

This paper addresses an important problem, and the method is novel.  Substantial analysis has been provided and all reviewers think the theoretical contribution is sufficient.  Although improvements are still possible in empirical evaluation especially against more state-of-the-art baseline, the consensus is that this paper makes a solid contribution to the proceeding.

**Additional Comments On Reviewer Discussion:**

The rebuttal has been noted by the reviewers and have been taken into account by the AC in the recommendation of acceptance/rejection.

---

### Decision · Program_Chairs · 2025-01-22

Accept (Poster)